# Regulation of centrosome size by the cell-cycle oscillator in *Drosophila* embryos

Siu-Shing Wong ⓘD, Alan Wainman ⓘD, Saroj Saurya ⓘD & Jordan W Raff ⓘD ✉

## Abstract

**Mitotic centrosomes assemble when centrioles recruit large amounts of pericentriolar material (PCM) around themselves. In early *C. elegans* embryos, mitotic centrosome size appears to be set by the limiting amount of a key component. In *Drosophila* syncytial embryos, thousands of mitotic centrosomes are assembled as the embryo proceeds through 13 rounds of rapid nuclear division, driven by a core cell cycle oscillator. These divisions slow during nuclear cycles 11–13, and we find that centrosomes respond by reciprocally decreasing their growth rate, but increasing their growth period—so that they grow to a relatively consistent size at each cycle. At the start of each cycle, moderate CCO activity initially promotes centrosome growth, in part by stimulating Polo/PLK1 recruitment to centrosomes. Later in each cycle, high CCO activity inhibits centrosome growth by suppressing the centrosomal recruitment and/or maintenance of centrosome proteins. Thus, in fly embryos, mitotic centrosome size appears to be regulated predominantly by the core cell cycle oscillator, rather than by the depletion of a limiting component.**

**Keywords** Centriole; Centrosome; Organelle Size; PCM; Cell Cycle
**Subject Categories** Cell Adhesion, Polarity & Cytoskeleton; Cell Cycle; Development

## Introduction

Centrosomes are important organisers of the cell that are formed when mother centrioles recruit pericentriolar material (PCM) around themselves (Conduit et al, 2015; Bornens, 2021; Vasquez-Limeta and Loncarek, 2021; Lee et al, 2021; Woodruff, 2021). The PCM consists of several hundred proteins (Alves-Cruzeiro et al, 2013), including many that help nucleate and organise microtubules (MTs), as well as many signalling molecules, cell cycle regulators, and checkpoint proteins—allowing the centrosome to function as both a major MT-organising centre and an important coordination centre in many eukaryotic cells (Arquint et al, 2014; Chavali et al, 2014).

In interphase, the centrosomes in most cells organise relatively little PCM, but in virtually all cells with centrosomes there is a dramatic increase in PCM recruitment as cells prepare to enter mitosis—a process termed centrosome maturation (Palazzo et al, 2000; Conduit et al, 2015; Vasquez-Limeta and Loncarek, 2021). The mitotic protein kinase Polo/PLK1 (fly/human nomenclature) plays a particularly important part in centrosome maturation (Lane and Nigg, 1996; Dobbelaere et al, 2008; Haren et al, 2009; Lee and Rhee, 2011; Conduit et al, 2014a; Woodruff et al, 2015b; Ohta et al, 2021), and the conserved Spd-2/CEP192 family of proteins help recruit Polo/PLK1 to mitotic centrosomes (Joukov et al, 2014; Meng et al, 2015; Decker et al, 2011; Alvarez Rodrigo et al, 2019; Ohta et al, 2021; Wong et al, 2022). In flies and worms, the centrosomal Polo/PLK1 recruited by Spd-2/SPD-2 phosphorylates the large coiled-coil protein Cnn (flies) or SPD-5 (worms), which then assembles into large macromolecular "scaffold" structures (Conduit et al, 2014a; Woodruff et al, 2015a; Feng et al, 2017; Woodruff et al, 2017; Cabral et al, 2019). These scaffolds give mechanical strength to the mitotic PCM (Lucas and Raff, 2007; Mittasch et al, 2020) and also help to recruit other PCM "client" proteins to the assembling mitotic centrosome (Woodruff et al, 2014; Raff, 2019).

How mitotic centrosome growth is regulated in somatic cells is unclear but, in early *C. elegans* embryos, centrosome size appears to be set by a limiting pool of SPD-2/CEP192 (Decker et al, 2011; Zwicker et al, 2014). The total embryonic pool of SPD-2 is thought to remain constant throughout the early cell division cycles, and it gets divided equally amongst the exponentially increasing number of centrosomes. As a result, the centrosomes halve in size after each round of cell division. A "limiting pool" of an organelle building block is potentially a powerful way to regulate organelle size, as it allows size to be set without the need for a specific size-measuring mechanism (Marshall, 2008; Goehring and Hyman, 2012; Marshall, 2020). It is unclear, however, if such a mechanism sets centrosome size in other cell types.

As described above, in flies Spd-2, Polo and Cnn, cooperate to guide the assembly of the mitotic PCM scaffold. Although Polo is better described as a "regulator" of scaffold assembly, for ease of presentation, we hereafter refer to these proteins collectively as "scaffold" proteins, to distinguish them from PCM "client" proteins that interact with the scaffold but are not essential for scaffold assembly. We recently developed methods to measure the growth kinetics of Spd-2, Polo and Cnn in living *Drosophila* syncytial blastoderm embryos—where we can simultaneously track hundreds of mitotic centrosomes as they rapidly and near-synchronously assemble during S-phase in preparation for mitosis (which, in these embryos, directly follows S-phase without any Gap phase)

Sir William Dunn School of Pathology, University of Oxford, Oxford OX1 3RE, UK. ✉E-mail: jordan.raff@path.ox.ac.uk

(Aydogan et al, 2018; Wong et al, 2022). The centrosomal levels of Polo, Spd-2 and Cnn all started to increase at the start of S-phase, as centrosomes started to mature in preparation for mitosis; but whereas Cnn levels continued to rise and/or plateau as the embryos entered mitosis, the centrosomal levels of Polo and Spd-2 started to decrease before the entry into mitosis (Wong et al, 2022) (Fig. 1A,B). Thus, the proteins required for mitotic PCM scaffold assembly exhibit different growth kinetics, making it hard to use these proteins to define centrosome "size" at any particular point in the cell cycle. We wondered, therefore, whether measuring the growth kinetics of PCM-client proteins might provide more insights into how centrosome size is regulated in these embryos.

Here we examine the recruitment kinetics of 7 PCM-client proteins. We find that their centrosomal levels increase during S-phase, reach maximal levels just prior to mitosis, before decreasing as the embryos enter mitosis. Unlike in early worm embryos, the mitotic centrosomes in these fly embryos grow to a relatively consistent maximal size during nuclear cycles (NC) 11, 12 and 13. This consistency of size seems to arise because there is an inverse relationship between the centrosome growth rate and growth period. In NC11, S-phase is short and the centrosomes grow quickly for a short time; in subsequent cycles, S-phase is longer and the centrosomes grow more slowly but for a longer time. In this way, the centrosomes grow to a similar size irrespective of nuclear cycle length or the number of centrosomes in the embryo. We find that the core cell cycle oscillator (CCO) plays an important part in setting centrosome size in these embryos, as it reciprocally influences the rate and period of mitotic centrosome growth.

## Results

### The dynamics of mitotic centrosome growth in the early *Drosophila* embryo

Due to their rapid nuclear cycle timing, the mitotic centrosomes in early *Drosophila* embryos start to grow at the start of S-phase, in preparation for the next round of mitosis. We previously showed that the recruitment dynamics of Spd-2, Polo and Cnn (that cooperate to build a mitotic PCM scaffold) were surprisingly complicated (Wong et al, 2022) (Fig. 1A,B). To examine how PCM-client proteins were recruited to the PCM scaffold, we quantified the centrosomal fluorescence levels of γ-tubulin-GFP, Msps-GFP, GFP-TACC, Aurora A-GFP and the γ-tubulin-ring complex (γ-TuRC)-associated proteins Grip71-GFP, Grip75-GFP and Grip128-GFP in living embryos as they proceeded through NC11-13 (Fig. 1A,C,D). In these experiments, we define centrosome "size" by the amount of protein recruited, measured by centrosomal fluorescence intensity. We obtained similar results, however, if we used centrosome area (measured from 2D projections of Z-stacks through the entire centrosome volume) as a measure of centrosome size (Fig. EV1).

In all our experiments, we defined mitosis as starting at nuclear envelope breakdown (NEB) and lasting until the centrosomes first detectably separated, indicating the start of S-phase. The centrosomal levels of all seven PCM-client proteins increased during S-phase, but started to decline rapidly shortly before the embryos entered mitosis (mitosis is indicated by the grey areas on the graphs in Fig. 1B–D). Intriguingly, the PCM-client recruitment profiles fell into two classes that were most clearly defined by their distinct behaviours during NC13. The centrosomal levels of TACC, Msps and γ-tubulin (Class I, Fig. 1C) tended to increase relatively steadily through most of NC13, but then exhibited a noticeable increase in their recruitment rate towards the end of S-phase, which was not observed for Aurora A or any of the γ-TuRC components (Class II, Fig. 1D). In flies, γ-tubulin exists in two complexes: the γ-TuRC and the γ-tubulin-small complex (γ-TuSC) (Oegema et al, 1999), presumably explaining why γ-tubulin and the γ-TuRC components can exhibit different recruitment kinetics. Importantly, these different dynamics were not due to differences in the promoters used to drive each protein's expression: if we normalised the incorporation profiles for S-phase length and fluorescence intensity then the proteins within each Class exhibited very similar recruitment kinetics irrespective of the promotors used to drive their expression (Appendix Fig. S1). Thus, PCM-client proteins appear to be recruited to mitotic centrosomes in at least two different ways.

The decline in the centrosomal levels of all the PCM-client proteins just prior to NEB was unexpected, as we presumed that centrosomes would maximise their MT-organising capacity after NEB, when the mitotic spindle is being assembled. To examine whether this decline in PCM was accompanied by a decline in centrosomal MTs, we measured MT fluorescence intensity in the area occupied by the centrosomes in embryos expressing Spd-2-mCherry (as a centrosome marker) and Jupiter-GFP (as a MT marker) (Karpova et al, 2006) (Fig. 2). In NC11-12, centrosomal MT levels increased in early S-phase, dipped in ~mid-S-phase, increased rapidly prior to NEB, before dropping sharply again at NEB. Centrosomal MT behaviour during NC13 was more complicated in the early S-phase (perhaps because this data is aligned to NEB [$t = 0$], so the data at the start of the long S-phase of NC13 is less well-synchronised), but the same trend was observed around the entry into mitosis. We suspect that this behaviour is driven by changes in CCO activity, and may reflect the centrosomal MTs being organised by different combinations of scaffold and Class I and Class II client proteins during the cycle. We conclude that the amount of both PCM and MTs organised by centrosomes decreases as these embryos enter mitosis proper. The reasons why this may be beneficial in these embryos are considered in "Discussion".

### Centrosome growth rate and growth period are inversely correlated so that centrosomes grow to a similar size at NC11-13

To quantify and compare the growth parameters of all the PCM scaffold and PCM-client proteins, we calculated each protein's average initial and peak centrosome-fluorescence intensity and their average rate and period of growth at each division cycle (Fig. 3). Strikingly, centrosomes generally grew to a similar maximum size at each successive cycle ("peak intensity" graphs in green boxes, Fig. 3), and this was also observed when we used centrosome area as a measure of size ("Peak area" graphs in green boxes, Fig. EV2). For some centrosomal proteins, there was a downward trend in their maximal fluorescence intensity across successive cycles (most prominently Grip75-GFP), but this decrease was usually relatively modest and nowhere near the ~50% decrease observed after each cell division in early *C. elegans* embryos

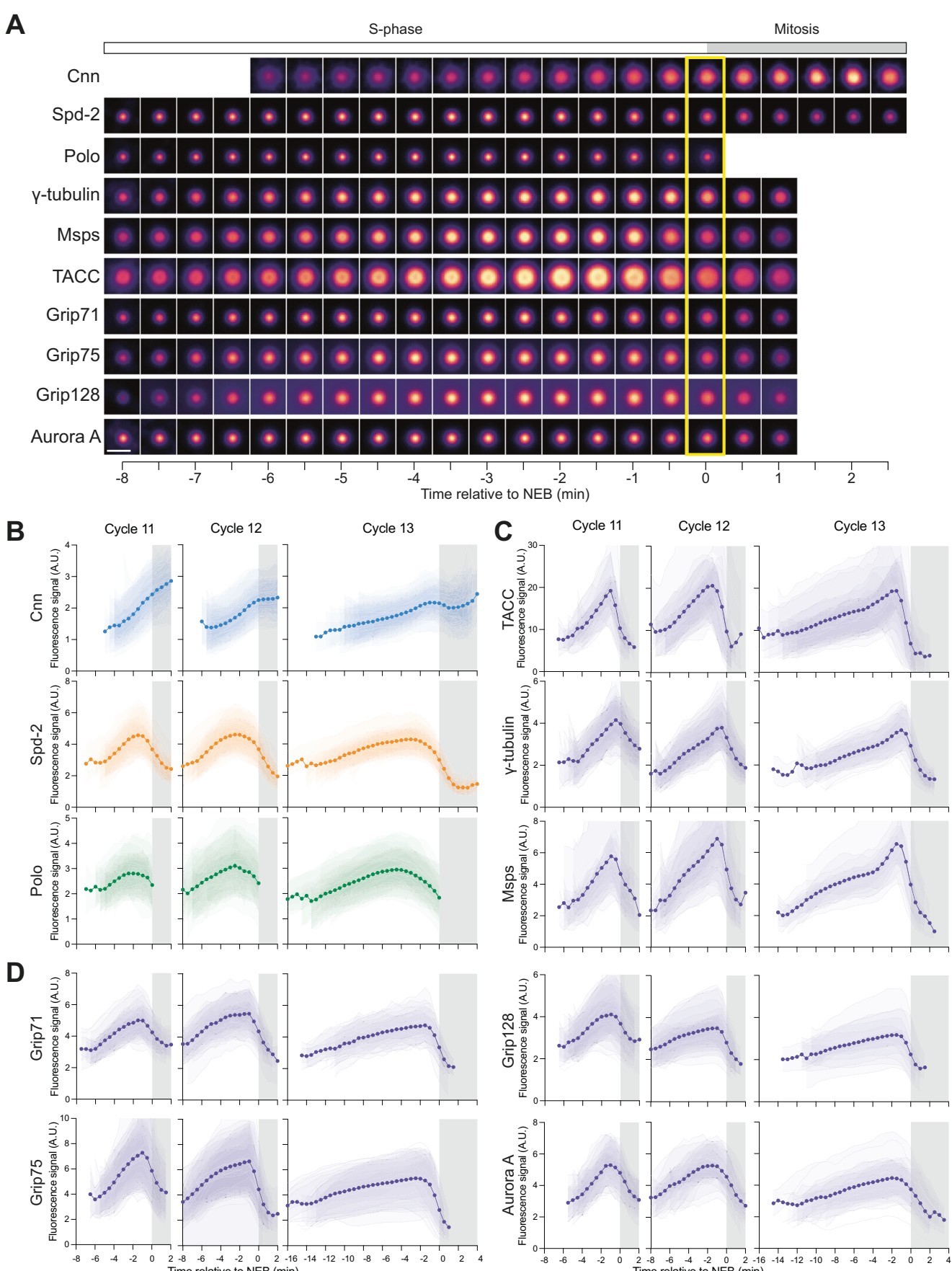

**Figure 1. Centrosome growth kinetics in the *Drosophila* syncytial blastoderm embryo.**

(A) Images show how the centrosomal fluorescence intensity of several centrosome proteins varies during NC12 in a representative embryo ($t = 0 =$ NEB, indicated by yellow box). The images were obtained by averaging the fluorescence intensity distribution of all of the centrosomes in a single embryo at each timepoint. Note that for technical reasons not all of the centrosome proteins can be followed for the full time period (see "Methods"). Scale bar = 2 μm. (B–D) Graphs show how the mean centrosomal fluorescence intensity (±SD of the data in each individual embryo shown in reduced opacity) of the PCM scaffold proteins Cnn, Spd-2 and Polo (B), and the Class I (C) and Class II (D) PCM-client proteins varies during NC11, 12 and 13. All individual embryo tracks were aligned to NEB ($t = 0$). The white parts of the graphs represent S-phase, and the grey parts represent mitosis. $N =$ 7–15 embryos analysed at each nuclear cycle for each marker with a total of $n = \sim$200–400, ~400–800 or ~600–1200 total centrosomes analysed at NC11, 12 and 13, respectively. Note that the data for the PCM scaffold proteins (B) was shown previously (Wong et al, 2022), but is reproduced here to allow comparison to the PCM-client proteins, as these datasets were all acquired during a contemporaneous period.

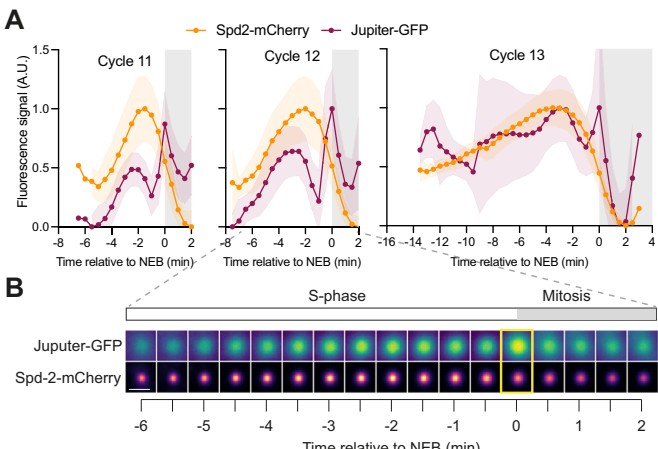

**B**

**Figure 2. A comparison of centrosome and centrosome-MT growth kinetics in the *Drosophila* syncytial blastoderm embryo.**

(A, B) Graphs (A) and images from a representative embryo during NC12 (B) show how the centrosomal fluorescence intensity (mean ± SD in the graphs) of Spd-2-mCherry (orange) and of the centrosomal MTs (monitored with the MT-binding protein Jupiter-GFP, purple) change over time during NC11-13. Data were obtained simultaneously from embryos co-expressing both proteins. The fluorescence intensity of the centrosomal MTs was monitored by measuring fluorescence intensity in a tightly cropped region of interest surrounding the centrosomal Spd-2 signal. Images were obtained by averaging the fluorescence intensity distribution of all of the centrosomes in a single embryo at each timepoint. All individual embryo tracks were aligned to NEB ($t = 0$, yellow box in images). The white parts of the graphs represent S-phase, and the grey parts represent mitosis. $N = 11$ embryos analysed at each nuclear cycle for each marker with a total of $n = \sim$300–600, ~700–1200 or ~1000–2000 total centrosomes analysed at NC11, 12 and 13, respectively. Scale bar = 2 μm.

(Decker et al, 2011). We conclude that mitotic centrosomes generally grow to a similar, or only slightly smaller, maximal size during NC11-13. Interestingly, the size of the metaphase spindles and of the nuclei decreased significantly during NC11-13 (Fig. EV3), indicating that centrosome size and spindle/nuclear size do not scale proportionally in these embryos (see "Discussion").

Although each centrosome protein grew to a relatively consistent maximal size during NC11-13, their growth rate and growth period changed significantly at each cycle. In general, as S-phase length increased, the centrosome growth period also increased (graphs in yellow boxes, Fig. 3), but the centrosome growth rate decreased (graphs in pink boxes, Fig. 3). This was also generally true if we used centrosome area as a measure of size (Fig. EV2). Thus, centrosomes appear to grow to a relatively consistent size during NC11-13 because there is an inverse relationship between the centrosome growth rate and growth period (Fig. EV4).

## Cdk/Cyclin activity influences the centrosome growth rate and growth period

We noticed that in general there was also an inverse correlation between S-phase length and the centrosome growth rate (as S-phase length increased, the centrosome growth rate decreased; middle graphs for each protein, Fig. EV4) and a strong linear correlation between S-phase length and the centrosome growth period (as S-phase length increased, the centrosome growth period also increased; right graphs for each protein, Fig. EV4). As S-phase length in these measurements is largely determined by the time it takes for Cdk/Cyclin activity to reach the threshold required to trigger NEB, this correlation suggests that Cdk/Cyclin activity influences both the centrosome growth rate and period. To directly test this possibility, we examined the recruitment dynamics of the PCM clients γ-tubulin and TACC in embryos in which we halved the genetic dose of *Cyclin B* (hereafter $CycB^{1/2}$ embryos). This perturbation slows the rate of Cdk/Cyclin activation so S-phase length increases, as it takes longer for the embryos to enter mitosis (Hayden et al, 2022; Aydogan et al, 2022). In $CycB^{1/2}$ embryos the growth period of both PCM proteins was increased, but their growth rate was slowed (Fig. 4). We conclude that Cdk/Cyclin activity can influence both the centrosome growth rate and growth period.

## Cdk/Cyclins appear to phosphorylate Spd-2 to reduce its centrosomal recruitment and/or maintenance during mitosis

How might Cdk/Cyclins influence the centrosome growth period? Perhaps the simplest hypothesis is that as embryos prepare to enter mitosis Cdk/Cyclins (or perhaps other mitotic kinases whose activity oscillate largely in phase with Cdk/Cyclins, such as Polo or Aurora A) phosphorylate one or more of the PCM scaffold and/or client proteins to decrease their centrosomal recruitment and/or maintenance. In such a scenario, a threshold level of CCO activity would help to "switch off" mitotic centrosome growth, so centrosomes would grow for a longer period in embryos where the rate of Cdk/Cyclin activation is slowed—as we observe when we reduce the dosage of Cyclin B, or as occurs naturally as S-phase slows during NC11-13 (Farrell and O'Farrell, 2014; Deneke et al, 2016).

As Cdk/Cyclins have a well-defined minimal consensus phosphorylation site sequence (Ser/Thr-Pro) we first tested whether phosphorylation by Cdk/Cyclins could potentially influence the behaviour of the scaffold proteins Spd-2 and/or Cnn by mutating these Ser/Thr-Pro motifs to Ala-Pro (generating Spd-2-Cdk20A and Cnn-Cdk6A—Appendix Fig. S2). We have previously shown that both proteins are phosphorylated specifically at centrosomes in embryos (Conduit et al, 2014b),

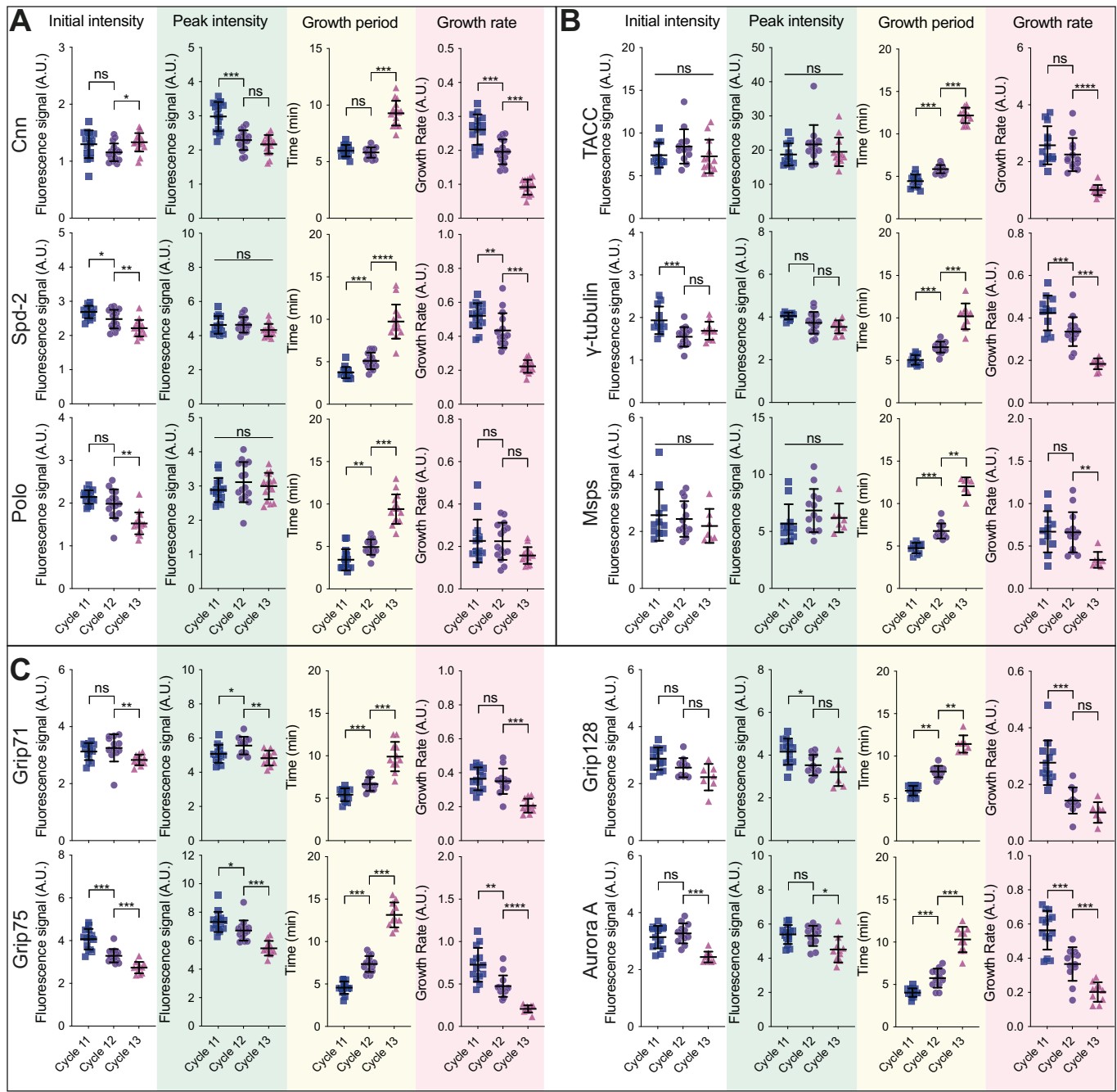

**Figure 3. Analysis of centrosome growth parameters during NC11, 12 and 13.**

(A–C) Scatter plots show the mean (±SD) initial fluorescent intensity (left graphs for each protein), peak fluorescent intensity (boxed in green), growth period (boxed in yellow), and growth rate (boxed in pink) of centrosomes during NC11, 12 or 13 for the PCM-scaffolding proteins (A) and the PCM-Class I (B) and Class II (C) client proteins. Each data point represents the average of all the centrosomes in an individual embryo (calculated from the data shown in Fig. 1). Statistical comparisons used either an ordinary one-way ANOVA (Gaussian-distributed and variance-equal), a one-way Welch ANOVA (Gaussian-distributed and variance-unequal), or a Kruskal–Wallis's test (non-Gaussian-distributed). If significant, multiple testing was performed using either Tukey-Kramer's test (Gaussian-distributed and variance-equal), Games–Howell's test (Gaussian-distributed and variance-unequal), or Mann–Whitney's $U$ test (non-Gaussian-distributed) (*$P < 0.05$, **$P < 0.01$, ***$P < 0.001$, ****$P < 0.0001$, ns not significant). Gaussian distribution was tested using D'Agnostino and Pearson's test. Variance homogeneity was tested using the Levene W test.

and previous Mass Spectroscopy screens in the embryos of several *Drosophila* species identified phosphorylation at 4 of the 6 potential sites we mutated in Cnn, and at 10 of the 20 potential sites we mutated in Spd-2 (Hu et al, 2019) (red and blue circles, Appendix Fig. S2). We

stress that we do not know if any of these potential sites in Spd-2 or Cnn are normally phosphorylated by Cdk/Cyclins, but, if any of them are, then this phosphorylation should be abolished by these mutations.

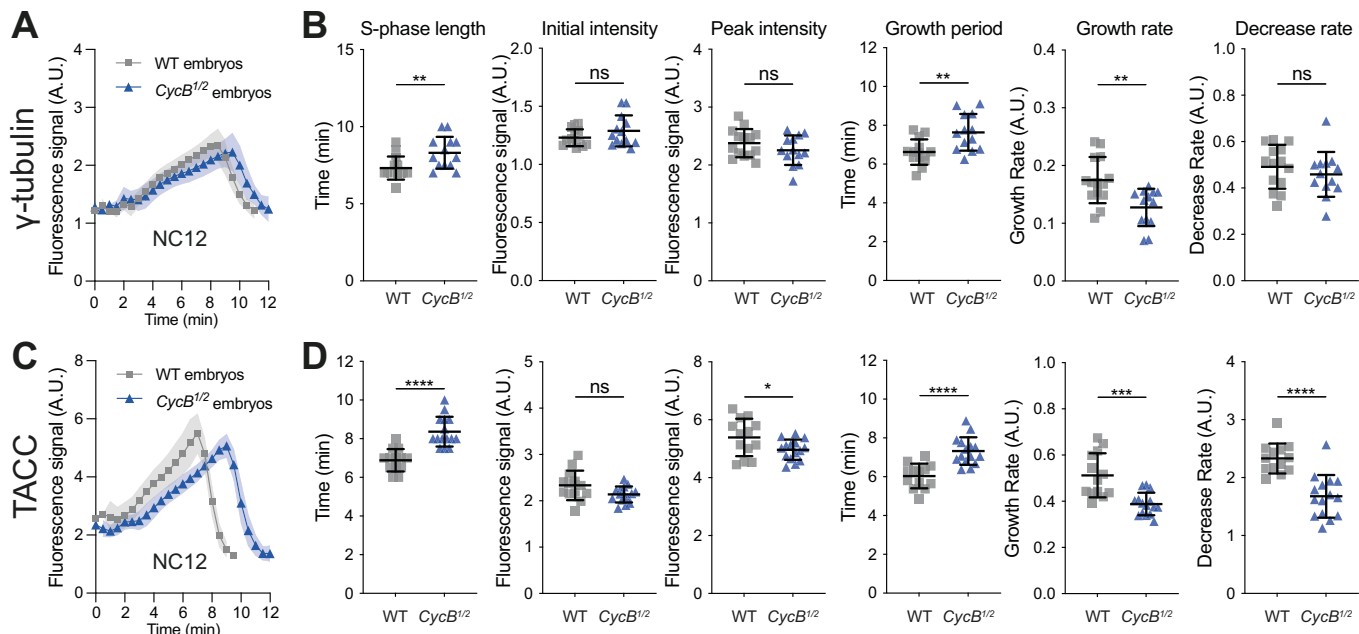

**Figure 4. Analysis of centrosomal growth kinetics in *CycB^{1/2}* embryos.**

(A, C) Graphs show how the centrosomal fluorescence intensity (mean ± SD) of γ-tubulin-GFP (A) or GFP-TACC (C) change over time during NC12 in embryos laid by either control wildtype females (*grey* lines) or heterozygous *CyclinB^{+/−}* females (*CycB^{1/2}* embryos) (*blue* lines). As S-phase length is extended in *CycB^{1/2}* embryos, these data were aligned to centrosome separation ($t = 0$), which occurs at the start of S-phase, rather than to NEB. $N = 10$–15 embryos and a total of $n = $ ~500–800 centrosomes were analysed for each condition. (B, D) Scatter plots compare various cell cycle and centrosome growth parameters in either WT (*grey*) or *CycB^{1/2}* embryos (*blue*). Statistical significance was assessed using an unpaired *t* test (*$P < 0.05$, **$P < 0.01$, ***$P < 0.001$, ****$P < 0.0001$, ns not significant). Note how, on average, S-phase was more extended in the *CycB^{1/2}* embryos expressing GFP-TACC compared to those expressing γ-tubulin-GFP, presumably helping to explain why the growth period and growth rate were more significantly perturbed in the GFP-TACC expressing embryos. The lower decrease rate observed in the *CycB^{1/2}* embryos expressing GFP-TACC may be due to this protein accumulating to slightly lower levels in the *CycB^{1/2}* embryos, and the disassembly rate being proportional to the amount of protein at the centrosome (the more protein is present, the faster it will disassemble).

We generated transgenic lines expressing either untagged or mNeonGreen (NG)-tagged versions of the mutant proteins, and found that all of these lines were dominant male sterile. The expression of these proteins in spermatocytes led to an accumulation of cytoplasmic aggregates during meiosis (Appendix Fig. S3) and a subsequent failure in cytokinesis, potentially explaining why these lines are male sterile. This dominant male sterility will be investigated elsewhere, but it meant that we could only examine the behaviour of the mutant proteins in embryos laid by females carrying one copy of the transgene and two WT (untagged) copies of the endogenous gene (i.e., in the presence of significant amounts of WT, untagged protein). As controls, we therefore examined the behaviour of WT Spd-2-NG and WT NG-Cnn in embryos expressing one copy of the NG fusion in the presence of two copies of the WT untagged endogenous gene (Fig. 5A,G). We compared the recruitment kinetics of the WT and mutant fusion proteins during NC12.

WT Spd-2-NG and Spd-2-Cdk20A-NG were present in embryos at similar levels (Fig. 5A), but the centrosomal levels of the mutant protein were elevated, its growth period was extended, and the rate at which its centrosomal levels decreased as the embryos entered mitosis was dramatically slowed (Fig. 5B–D). This decrease in the rate at which centrosomal Spd-2-Cdk20A levels dropped during mitosis meant that the mutant protein was still present at centrosomes at elevated levels at the start of the next cycle (probably explaining why Spd-2-Cdk20A centrosomal levels are also too high at the start of S-phase). The Spd-2-

Cdk20A growth rate also appeared dramatically slowed, but this is because the centrosomes start each cycle with more Spd-2-Cdk20A, and the protein is then incorporated over a longer period, so reducing the average growth rate.

These observations are consistent with the possibility that Cdk/Cyclins normally phosphorylate Spd-2 to help reduce its centrosomal accumulation from ~mid-S-phase onwards. We cannot rule out, however, that these changes are due to unknown defects caused by the Ser/Thr-Ala substitutions. As a preliminary test of this alternative possibility, we injected mRNA encoding either WT Spd-2-NG, Spd-2-Cdk20A-NG or a potentially phospho-mimicking Spd-2-Cdk20E-NG into early embryos. In these experiments, the mRNA is quickly translated into protein that competes for binding to the centrosome with the endogenous unlabelled protein (Novak et al, 2016). We then measured the fluorescence intensity of the centrosomes in each injected embryo in ~mid-S-phase (at the Spd-2 peak) ~1 h after mRNA injection. This analysis confirmed that the Cdk20A mutant accumulated at centrosomes to higher levels than WT, and revealed that the Cdk20E mutant accumulated to slightly lower levels (Fig. EV5). This reciprocal behaviour of the Cdk20A and Cdk20E mutants indicates that a difference in their phosphorylation potential is a plausible explanation for their different behaviours.

As an alternative way to test our hypothesis that Cdk/Cyclins can phosphorylate Spd-2 to help inhibit its accumulation at centrosomes, we analysed the behaviour of a WT Spd-2-NG

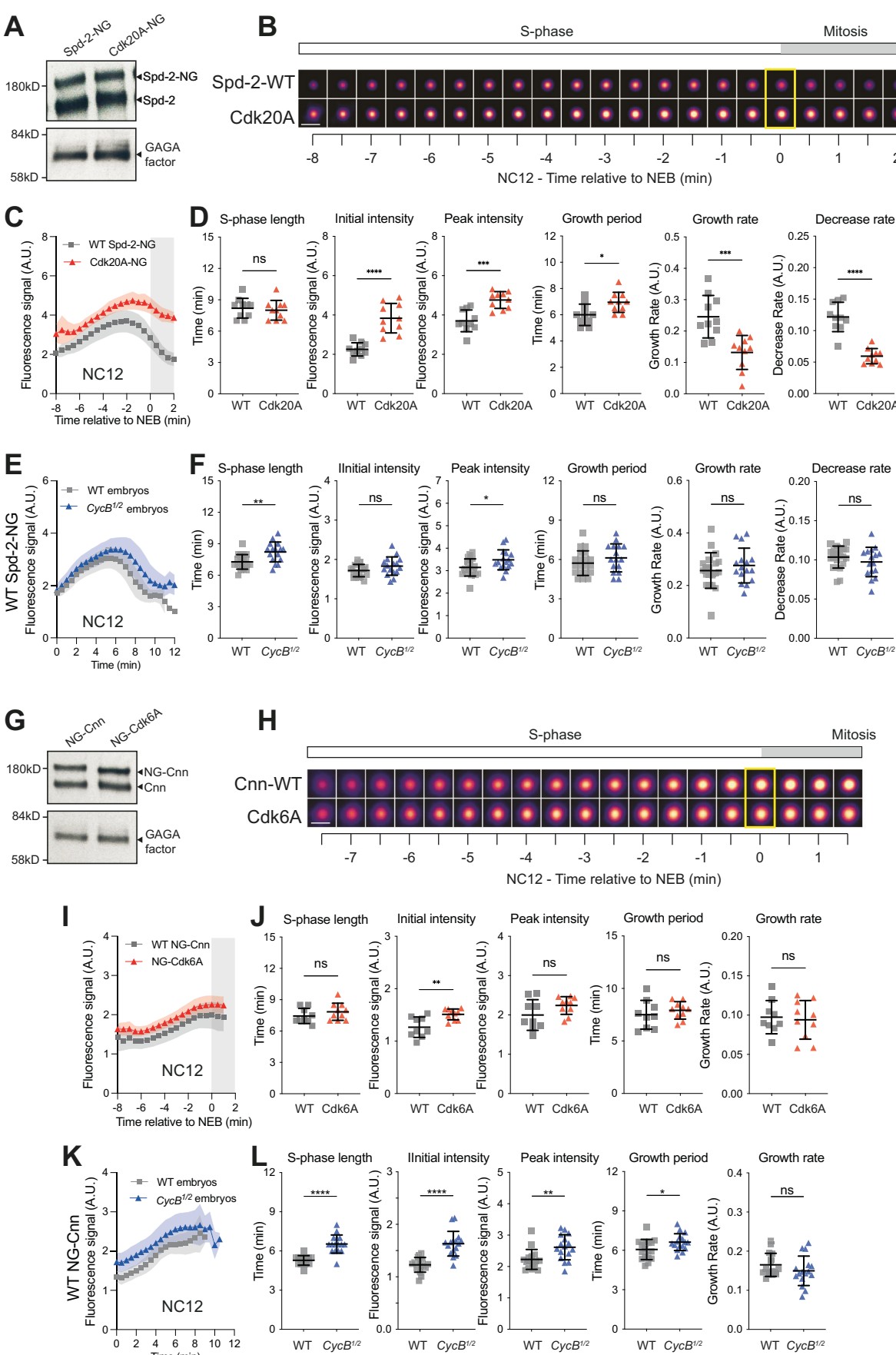

**Figure 5.   Analysis of centrosomal Spd-2 and Cnn growth kinetics when their potential phosphorylation by Cdk/Cyclins is perturbed.**

(**A**) Western blot shows protein levels of the endogenous Spd-2 and transgenically expressed WT Spd-2-NG or the Spd-2-Cdk20A mutant in early embryos. GAGA factor serves as a loading control. (**B, C**) Images of centrosomes from representative embryos (**B**) and a graph (**C**) comparing the centrosomal fluorescence intensity (mean ± SD in the graph) of WT Spd-2-NG or Spd-2-Cdk20A over time during NC12. Images were obtained by averaging the fluorescence intensity of all of the centrosomes in a single embryo at each timepoint. Individual embryo tracks were aligned to NEB ($t = 0$, yellow box on images); white parts of the graphs represent S-phase, grey parts mitosis. Scale bar = 2 μm. (**D**) Scatter plots show the mean (±SD) of various cell cycle and centrosome growth parameters derived from the data shown in (**C**). (**E**) Graph compares how the centrosomal fluorescence intensity (mean ± SD) of WT Spd-2-NG changes over time in either WT or $CycB^{1/2}$ embryos. (**F**) Scatter plots show the mean (±SD) of various cell cycle and centrosome growth parameters derived from the data shown in (**E**). (**G–L**) Panels show the same analyses as described in (**A–F**) but comparing the behaviour of WT NG-Cnn to the NG-Cnn-Cdk6A mutant. Note that the Cnn-fusion proteins tend to plateau in mitosis, so we did not calculate a decrease rate. $N = 9$–17 embryos and a total of $n = {\sim}400$–1000 centrosomes were analysed for each condition. Statistical comparisons were performed as described in the legend to Fig. 3.

transgene in $CycB^{1/2}$ embryos. In these embryos, it takes longer for CCO activity to reach peak levels (Hayden et al, 2022), so we would predict that WT Spd-2-NG should be recruited to centrosomes for a slightly longer period and so to slightly elevated levels, which is what we observed (Fig. 5E,F). This perturbation was subtle when compared to that seen with Spd-2-Cdk20A (Fig. 5C,D), but this is expected, as it takes only slightly longer for Cdk/Cyclin activity to reach the threshold required to inhibit WT Spd-2 accumulation in the $CycB^{1/2}$ embryos, whereas the centrosomal accumulation of Spd-2-Cdk20A is presumably not inhibited by Cdk/Cyclins at all. Moreover, we observed a similar difference in the behaviour of a Spd-2-NG fusion generated by CRISPR-mediated knock-in at the endogenous locus in WT and $CycB^{1/2}$ embryos (Appendix Fig. S4).

WT NG-Cnn and NG-Cnn-Cdk6A were present in embryos at similar levels (Fig. 5G), and the centrosomal levels of the mutant protein increased slightly compared to WT NG-Cnn (Fig. 5H–J). This difference was not nearly as dramatic as that seen for the Spd-2-Cdk20A-NG mutant (compare Fig. 5C and I), and most growth parameters of NG-Cnn-Cdk6A were not statistically different to WT (Fig. 5J). We also observed a modest increase in the centrosomal levels of WT NG-Cnn in $CycB^{1/2}$ embryos (Fig. 5K,L), but this could be explained, at least in part, by the increase in centrosomal Spd-2 levels in $CycB^{1/2}$ embryos, as Spd-2 helps recruit Cnn to centrosomes (Conduit et al, 2014b). Thus, any potential phosphorylation of Cnn by Cdk/Cyclins does not seem to strongly influence centrosomal Cnn recruitment and/or maintenance.

## Cdk/Cyclins appear to phosphorylate Spd-2 and Cnn to reduce their ability to recruit and/or maintain γ-tubulin at centrosomes

We next wanted to examine whether preventing the Cdk/Cyclin-dependent phosphorylation of Spd-2 and/or Cnn could influence the centrosomal recruitment of the PCM clients they help to recruit. We initially focused on γ-tubulin, because the Spd-2/CEP192 and Cnn/CDK5RAP2 family of proteins have both been implicated in recruiting γ-tubulin to centrosomes (Gomez-Ferreria et al, 2007; Zhu et al, 2008; Fong et al, 2008; Choi et al, 2010; Ohta et al, 2021; Tovey et al, 2021; Zhu et al, 2023). To confirm that this was also the case in fly embryos, we examined γ-tubulin recruitment in embryos laid by heterozygous $Spd$-$2^{+/-}$ or $cnn^{+/-}$ mutant mothers ($Spd$-$2^{1/2}$ and $Cnn^{1/2}$ embryos, respectively). Importantly, we previously showed that the centrosomal levels of Cnn are sensitive to its genetic dosage (Conduit et al, 2010), and found that this was also the case for Spd-2 (Appendix Fig. S5). Intriguingly, lowering Spd-2 levels reduced γ-tubulin accumulation throughout S-phase, whereas lowering Cnn levels perturbed γ-tubulin accumulation only in late S-phase (Fig. 6), indicating that

Spd-2 and Cnn help recruit γ-tubulin to centrosomes in different ways, as also reported recently (Zhu et al, 2023).

We next expressed a γ-tubulin-GFP fusion protein in embryos co-expressing untagged versions of either WT Spd-2 or WT-Cnn or the Spd-2-Cdk20A or Cnn-Cdk6A mutants. Compared to WT, the expression of the mutant proteins did not dramatically perturb the centrosomal recruitment of γ-tubulin-GFP (Fig. 7A), but the rate at which γ-tubulin-GFP left the centrosome as the embryos entered mitosis was reduced in both mutants (right graphs, Fig. 7B). Although this phenotype was subtle, it was statistically significant in both mutants ($P < 0.01$), and the presence of substantial amounts of WT Spd-2 and Cnn in these embryos (Fig. 5A,G) is likely to mask the potential severity of this phenotype. These observations suggest that Cdk/Cyclins normally phosphorylate Spd-2 and Cnn to decrease their ability to recruit and/or maintain centrosomal γ-tubulin as the embryos enter mitosis (explaining why γ-tubulin does not leave the centrosome as quickly as usual in the presence of the mutant proteins).

Finally, we wanted to examine whether mutating the potential Cdk/Cyclin phosphorylation sites in γ-tubulin (seven in total) perturbed its centrosomal recruitment dynamics. Unfortunately, a NG fusion to this mutant protein did not detectably localise to centrosomes, so we were unable to test this possibility.

## Cdk/Cyclin activity influences the centrosomal recruitment of Polo

Our studies so far suggest that rising levels of CCO activity help to switch off centrosome growth towards the end of each nuclear cycle, so explaining, at least in part, how the CCO might influence the centrosome growth period. But this mechanism cannot explain how the CCO influences the centrosome growth rate—i.e., why centrosomes tend to grow progressively more slowly during S-phase at successive nuclear cycles (pink graphs, Figs. 3 and EV2).

We noticed that at the start of NC11-13 the initial centrosomal recruitment of Polo-GFP—and also Spd-2-GFP, whose initial recruitment to centrioles/centrosomes appears to depend on Polo (Alvarez Rodrigo et al, 2021; Wong et al, 2022)—decreased at successive cycles, a trend that was not observed with most of the other centrosome proteins (left graphs for each individual protein, Fig. 3). As Polo/PLK1 is a major driver of mitotic centrosome growth (Sunkel and Glover, 1988; Lane and Nigg, 1996; Kalous and Aleshkina, 2023), we wondered whether Cdk/Cyclin activity might influence the centrosome growth rate by influencing the recruitment of Polo to centrosomes. To test this possibility, we examined centrosomal Polo-GFP dynamics in $CycB^{1/2}$ embryos during NC12 (Fig. 8). Strikingly, in these embryos, the initial recruitment of Polo-GFP to centrosomes at the start of S-phase was reduced, and there was a dramatic reduction in the rate of Polo-GFP recruitment,

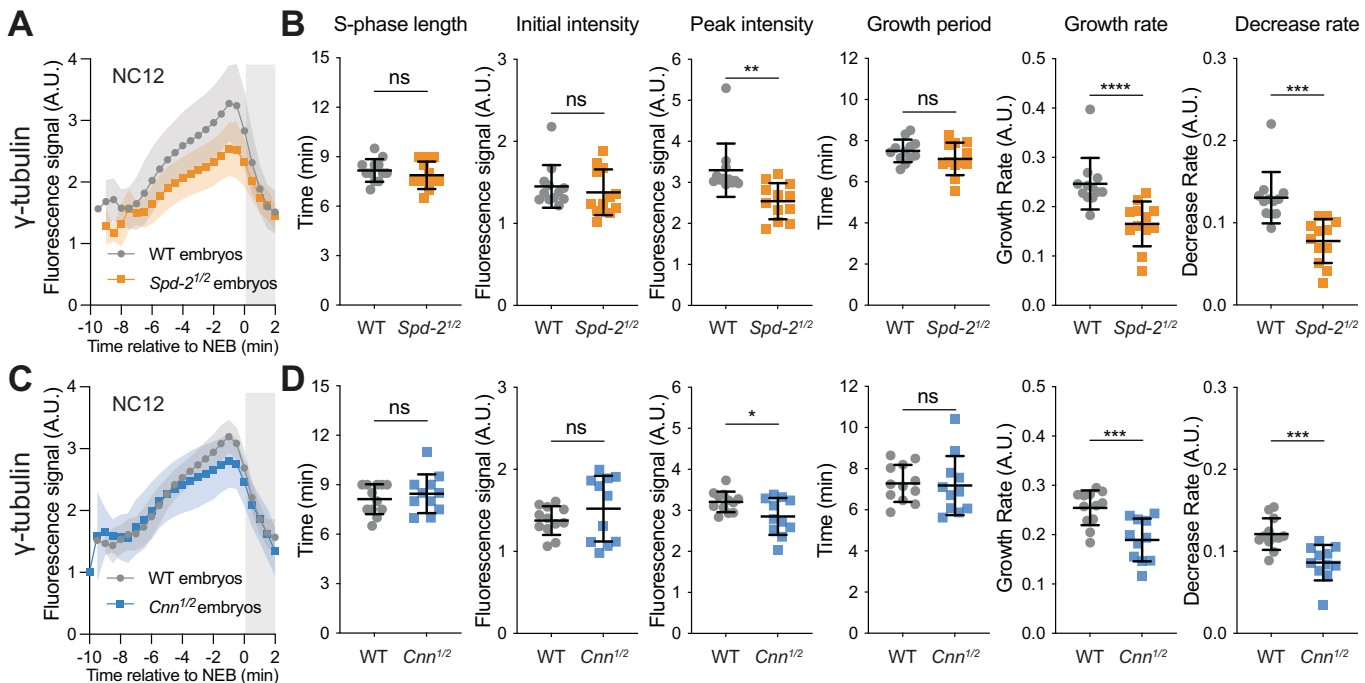

**Figure 6. Analysis of centrosomal γ-tubulin-GFP growth kinetics in embryos with reduced levels of Spd-2 or Cnn.**

(A, C) Graphs compare how the mean (±SD) centrosomal fluorescence intensity of γ-tubulin-GFP changes over time during NC12 in embryos laid by either WT mothers or mothers heterozygous for *Spd-2* or *cnn* mutations (*Spd-2^{1/2}* or *Cnn^{1/2}* embryos, respectively). Individual embryo tracks were aligned to NEB ($t = 0$); white parts of the graphs represent S-phase, and the grey parts mitosis. $N = 10$–15 embryos and a total of $n = $ ~500–800 centrosomes were analysed for each condition. (B, D) Scatter plots compare the mean (±SD) of various cell cycle and centrosome growth parameters in WT and *Spd-2^{1/2}* or *Cnn^{1/2}* embryos. Statistical comparisons were performed as described in the legend to Fig. 3. Note that the lower decrease rate observed in the half-dose embryos is likely due to the γ-tubulin accumulating to lower levels in the half-dose embryos, and the disassembly rate being proportional to the amount of protein at the centrosome (the more protein is present, the faster it will disassemble).

so that much less Polo was recruited to centrosomes during the nuclear cycle. We conclude that CCO activity normally promotes Polo recruitment to and/or maintenance at centrosomes, potentially helping to explain, at least in part, how CCO activity influences the centrosome growth rate.

## Discussion

Here we examine how mitotic centrosome size is regulated in *Drosophila* syncytial blastoderm embryos. We find that centrosomes grow to a relatively consistent size at nuclear cycles 11–13 because there is an inverse relationship between the centrosome growth rate and growth period. Centrosomes grow rapidly, but for a short period, in cycle 11, and then more slowly, but for a longer period, at subsequent cycles. As a result, centrosomes grow to a similar size no matter the length of the nuclear cycle or the number of centrosomes present in the embryo. This is in contrast to early worm embryos, where centrosomes halve in size at successive cell divisions, and this appears to be due to the depletion of a limiting component (SPD-2) (Decker et al, 2011). Given that fly and worm embryos use such a similar set of proteins to build their mitotic centrosomes (Conduit et al, 2015; Pintard and Bowerman, 2019), it is perhaps surprising that the mechanisms regulating centrosome size appear to be so different. Clearly, more work is required to resolve this paradox, but it is important

to note that centrosome size in the early *Drosophila* embryo is sensitive to the cytoplasmic levels of the core-scaffolding proteins Spd-2 and Cnn—as reducing their levels reduces centrosome size (Conduit et al, 2010) (Fig. 6; Appendix Fig. S5). It is just that in the fly system neither component seems to be sufficiently depleted from the cytoplasm to significantly limit centrosome growth as centrosome numbers increase. Instead, the activity of the Cdk/ Cyclin cell cycle oscillator (CCO) seems to regulate the kinetics of centrosome growth.

In early *Drosophila* embryos, the rate of CCO activation at the start of each nuclear cycle gradually slows during NC10-14 (Deneke et al, 2019; Farrell and O'Farrell, 2014). This leads to an increase in S-phase length and, as we show here, to a corresponding increase in the centrosome growth period and decrease in the centrosome growth rate. Halving the dosage of Cyclin B slows the centrosome growth rate, but increases the growth period, indicating that CCO activity directly influences both parameters. A priori, it seems likely that as each nuclear cycle progresses, the rising level of CCO activity influences centrosome growth in multiple ways that, at least in these embryos, are tuned to ensure that centrosomes grow to a relatively consistent, or only slightly smaller, size during NC11-13 (summarised in Fig. 9). This seems plausible as Cdk/Cyclins drive cell cycle progression, in part, by phosphorylating substrates at different critical activity thresholds (Coudreuse and Nurse, 2010; Swaffer et al, 2016). Moreover, Cdk/ Cyclins can influence biological processes by phosphorylating the

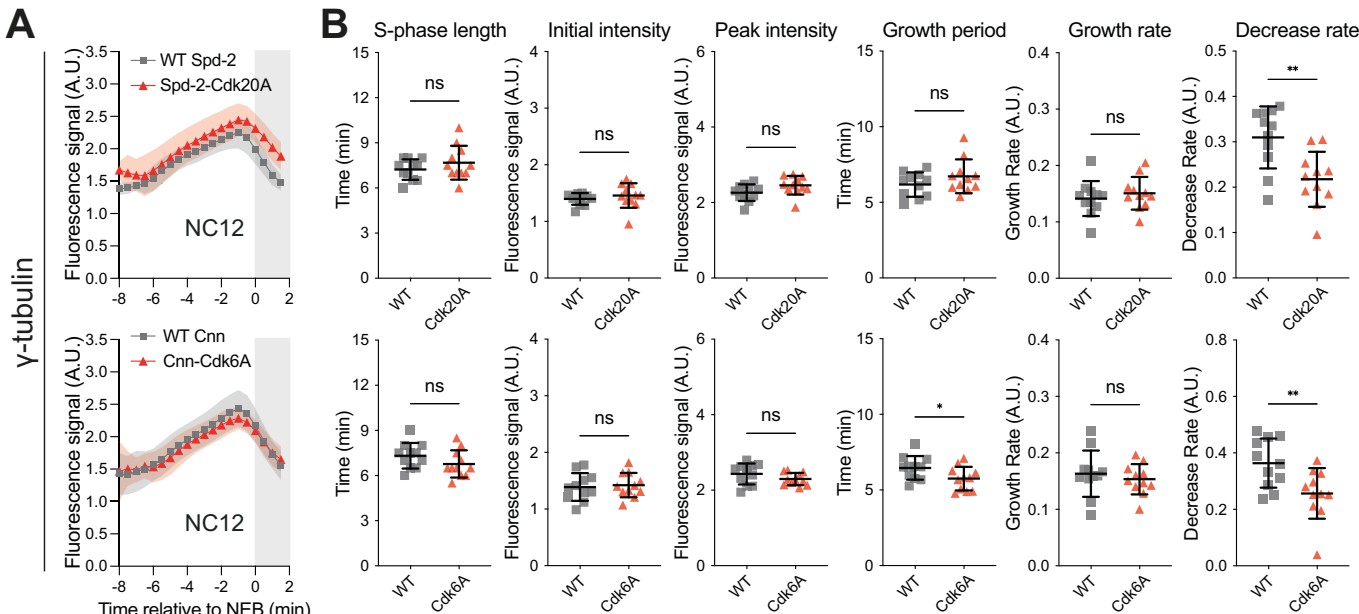

**Figure 7. Analysis of centrosomal γ-tubulin-GFP growth kinetics in embryos expressing mutant forms of Spd-2 or Cnn that should not be phosphorylated by Cdk/Cyclins.**

(A) Graphs compare how the mean centrosomal fluorescence intensity (±SD) of γ-tubulin-GFP changes over time during NC12 in WT embryos expressing untagged versions of either WT or mutant forms of Spd-2 (Spd-2-Cdk20A) (top graphs) or Cnn (Cnn-Cdk6A) (lower graphs). Individual embryo tracks were aligned to NEB (*t* = 0); white parts of the graphs represent S-phase, grey parts mitosis. In total, 10–15 embryos and a total of 500–800 centrosomes were analysed for each condition. (B) Scatter plots compare the mean (±SD) of various cell cycle and centrosome growth parameters in embryos expressing the WT and mutant proteins. Statistical comparisons were performed as described in the legend to Fig. 3.

same protein at different sites: some sites when Cdk/Cyclin activity is low, and others when it is high (Kõivomägi et al, 2011; Örd et al, 2019; Asfaha et al, 2022).

It is well-established that CCO activity initiates centrosome maturation, although it is largely unclear how it does so. In most somatic cells, centrosomes start to grow in G2, but in fly embryos CCO activity is already relatively high at the start of S-phase as this ensures that normally late-replicating DNA replication origins fire early so the genome can be duplicated quickly (Farrell and O'Farrell, 2014). This presumably explains why centrosomes can start to grow in preparation for mitosis at the start of S-phase in early *Drosophila* embryos. Based on our previous work (Wong et al, 2022), and our studies reported here, we propose that the CCO initiates centrosome growth, at least in part, by stimulating the centrosomal recruitment of Polo (Fig. 9(i)). Polo/PLK1 is recruited to centrosomes by binding to phosphorylated S–S/T(P) motifs, partially activating the kinase (Lee et al, 1998; Song et al, 2000; Elia et al, 2003; Reynolds and Ohkura, 2003; Liu et al, 2004). In fly embryos, S–S/T(P) motifs in the centriole proteins Ana1/CEP295 (Saurya et al, 2016) and Spd-2/CEP192 help recruit Polo first to centrioles (mainly dependent on Ana1) and then to the expanding mitotic PCM (mainly dependent on Spd-2) (Alvarez Rodrigo et al, 2019; Alvarez Rodrigo et al, 2021; Wong et al, 2022). The precise S–S/T(P) motifs required for Polo recruitment are not known, but several candidates are potential Cdk/Cyclin or PLK1 phosphorylation sites. Thus, CCO activity could promote Polo recruitment to centrioles/centrosomes by phosphorylating Ana1 and/or Spd-2 on S–S/T(P) sites.

As S-phase proceeds, CCO activity continues to rise (Deneke et al, 2016), and we propose that by mid-late S-phase a second activity threshold is crossed that triggers a second phase of phosphorylations (Fig. 9(ii–iii)). One consequence is that the centrosomal accumulation of Spd-2 starts to decrease at this stage, prior to NEB (Fig. 1A,B). This may be because the ability of the centrioles to recruit Polo to generate the Spd-2-scaffold starts to decrease (Wong et al, 2022), and the results we report here suggest that the direct phosphorylation of Spd-2 by Cdk/Cyclins may also play a part.

This second phase of phosphorylation may also help to explain why PCM clients are recruited to mitotic centrosomes in at least two different ways. Perhaps Class I (γ-tubulin, Msps and TACC) and Class II (Aurora A and the γ-TuRC components Grip71, Grip75 and Grip128) client proteins are initially recruited to mitotic centrosomes in a Spd-2-dependent manner, but the second phase of CCO-dependent phosphorylation promotes additional interactions of the Class I proteins with Cnn. In this way, the centrosomal accumulation pattern of Class II proteins would broadly mirror the pattern of Spd-2 (although they are not identical, perhaps because Spd-2 is recruited to both centrioles and to the PCM), while the Class I proteins exhibit an additional burst of Cnn-dependent recruitment shortly before the entry into mitosis. In support of this possibility, the centrosomal recruitment of γ-tubulin appears to be influenced by Spd-2 through most of S-phase, but by Cnn only in late S-phase (Fig. 6). Moreover, the phosphorylation of Cnn, and its worm equivalent SPD-5, by Polo/PLK1 has recently been shown to promote the centrosomal

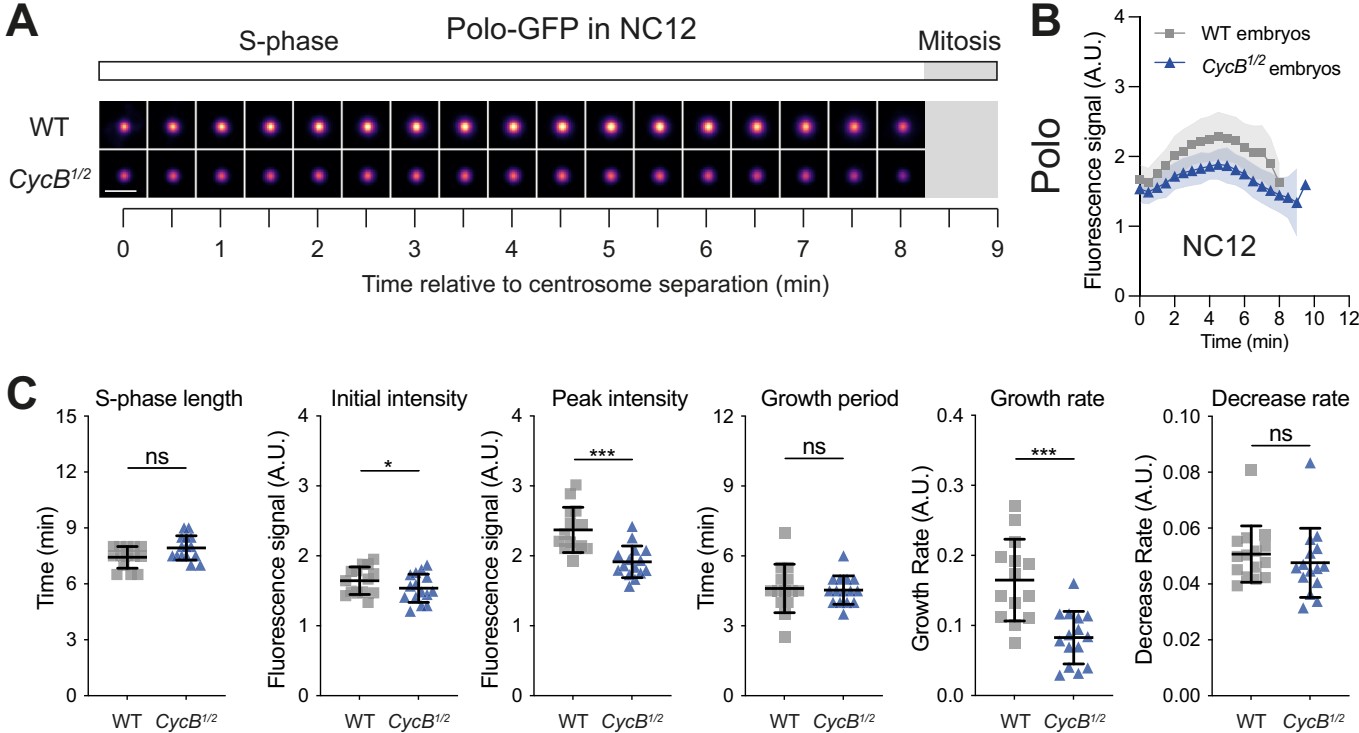

**Figure 8.   Analysis of Polo-GFP centrosome growth kinetics in *CycB*$^{1/2}$ embryos.**

(A, B) Images of centrosomes from representative embryos (A) and graphs (B) comparing how the centrosomal fluorescence intensity (mean ± SD in the graph) of Polo-GFP changes over time during NC12 in WT and *CycB*$^{1/2}$ embryos. Note that the time points shaded in grey indicate mitosis, which cannot be monitored in these embryos as Polo-GFP localises strongly to the mitotic kinetochores, making it difficult to track the centrosomes. The images were obtained by averaging the fluorescence intensity of all of the centrosomes in a single embryo at each timepoint. Scale bar = 2 µm. (C) Scatter plots show the mean (±SD) of various cell cycle and centrosome growth parameters derived from the data shown in (B). N = 14–15 embryos and a total of n = ~500–800 centrosomes were analysed for each condition. Statistical comparisons were performed as described in the legend to Fig. 3.

recruitment of specifically the small γ-TuSC in flies and worms (Ohta et al, 2021; Tovey et al, 2021)—potentially explaining why γ-tubulin, but none of the fly γ-TuRC components, exhibit a burst of recruitment prior to mitotic entry. These findings suggest that centrosome maturation in these embryos occurs in two phases: a first phase dominated by Spd-2-dependent recruitment, and a second phase dominated by Cnn-dependent recruitment.

Finally, we propose that, just prior to NEB, a third CCO-activity threshold is crossed that triggers the effective release of the PCM-client proteins from centrosomes (Fig. 9(iv)). Our data suggests that the phosphorylation of Spd-2 and Cnn by Cdk/Cyclins may play a part in this release, but we suspect that many additional phosphorylation events—promoted by Cdk/Cyclins or other components of the CCO (such as Polo and Aurora A) and acting on a combination of Cnn and/or Spd-2 and/or the PCM-client proteins—will contribute to this highly coordinated process. This rapid decline in PCM-client protein levels prior to NEB may be an unusual feature of the early fly embryo, as it is thought that centrosomes do not start to disassemble until the metaphase/anaphase transition in most cell types. Importantly, however, previous studies that have measured or inferred the kinetics of centrosome growth in vertebrate cells often indicate that mitotic centrosomes stop growing as cells enter mitosis (Piehl et al, 2004; Khodjakov and Rieder, 1999). Thus, in both fly embryos and vertebrate cells mitotic centrosomes stop growing during mitosis (although the centrosomes do not appear to lose their client proteins until later in mitosis in vertebrate cells). Thus, the ability of mitotic levels of CCO activity to inhibit mitotic centrosome growth may be a conserved feature of centrosome maturation in flies and vertebrates.

Why might mitotic centrosomes start to disassemble their Spd-2 scaffold and release their client proteins prior to NEB in early fly embryos? We suspect that this may be because there is very little time for mitotic centrosomes to disassemble at the end of mitosis in these rapidly cycling embryos, making it is necessary to start disassembling the mitotic PCM unusually early. This may also explain why, unlike worm embryos, fly embryos have strong non-centrosomal pathways of spindle assembly (Hayward et al, 2014). In fly embryos, centrosomal MTs might be most important for breaking down the nuclear envelope (Katsani et al, 2008), whereas, after NEB, it may be useful to divert MT nucleating/organising resources away from centrosomes to the other pathways of spindle assembly (Petry, 2016; Prosser and Pelletier, 2017). This may also help to explain our observation that centrosome size and spindle size do not scale proportionally in early fly embryos. Centrosomes may not be the dominant pathway of spindle assembly in fly embryos, whereas they appear to be in worm embryos, where centrosome size scales with spindle size (Greenan et al, 2010).

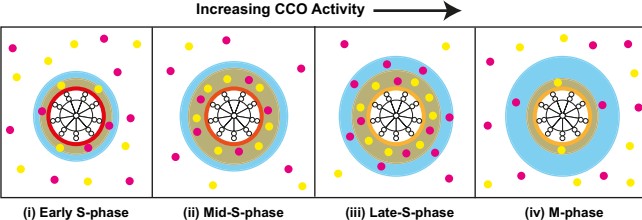

Increasing CCO Activity ⟶

(i) Early S-phase    (ii) Mid-S-phase    (iii) Late-S-phase    (iv) M-phase

**Figure 9. Schematic summary of how CCO activity might regulate mitotic centrosome growth in syncytial blastoderm *Drosophila* embryos.**

(i) In early S-phase embryos Cdk/Cyclin activity is high enough to activate the centrioles (indicated in red) to allow them to recruit Polo and initiate Spd-2/Cnn scaffold assembly. Mitotic centrosomes start to grow, driven primarily by the expansion of the scaffold, which recruits PCM-client proteins. Our data suggest that γ-tubulin is predominantly interacting with the Spd-2 scaffold at this stage, and this may be true of other client proteins. (ii–iii) By mid-late S-phase, the Spd-2 scaffold has reached its maximal size, as rising CCO activity helps to switch off the centrioles ability to recruit Polo to generate the Spd-2 scaffold (indicated in dark orange) (Wong et al, 2022), and, at this level of CCO activity, Spd-2 starts to be phosphorylated to further inhibit its centrosomal accumulation. This level of CCO activity also phosphorylates Cnn and/or the Class I client proteins (such as γ-tubulin) to stimulate their interactions, leading to a burst of additional Class I (but not Class II) client protein recruitment prior to mitotic entry. (iv) As the embryos enter mitosis the centrioles recruit very little Polo or Spd-2 (indicated by showing centrioles in light orange) and Spd-2 is fully phosphorylated by Cdk/Cyclins (and perhaps other CCO kinases) to help inhibit its centrosomal accumulation. Even though the Spd-2 scaffold is shrinking at this stage, the Cnn scaffold is maintained by the high levels of active Polo in the cytoplasm (and low levels of competing phosphatase), which prevent its disassembly. This high-level of mitotic CCO activity further phosphorylates the scaffold and/or the client proteins to inhibit their interactions; the client proteins are no longer recruited efficiently by the scaffold and so they start to leave.

# Methods

### *Drosophila melanogaster* stocks and husbandry

The *Drosophila* stocks used, generated and/or tested in this study are listed in Table 1. The precise stocks used in each experiment (and the relevant Figure) are listed in Table 2. Flies were maintained on *Drosophila* culture medium (0.68% agar, 2.5% yeast extract, 6.25% cornmeal, 3.75% molasses, 0.42% propionic acid, 0.14% tegosept and 0.7% ethanol) in 8-cm × 2.5-cm plastic vials or 0.25-pint plastic bottles. For microscopy and immunoblot experiments, flies were placed in embryo collection cages on fruit juice plates (see below) with a drop of yeast paste. Fly handling was performed as previously described (Roberts, 1998).

# Transgenic fly line generation

Transgenic fly lines were generated via random P-element insertion (injected, mapped, and balanced by "The University of Cambridge Department of Genetics Fly Facility"). For transgene selection, the $w^+$ gene marker was included in the transformation vectors, and constructs were injected into the $w^{1118}$ genetic background.

To generate Spd-2-Cdk20A mutants, two different cDNA fragments encoding the following amino acid substitutions: S49A; T112A; S311A; T337A; S484A; T516A; S531A, S536A; T561A; S606A; S614A; S618A; S625A; S944A; S1021A;T1023A; S1065A;

**Table 1. *Drosophila* stocks used in this study.**

| Allele | Source |
|---|---|
| WT, Canton S | Bloomington Stock Centre |
| cnn^f04547 | Exelixis stock no. f04547, Exelixis Stock Centre (Harvard Medical School, Boston, MA). |
| cnn^HK21 | (Megraw et al, 1999; Vaizel-Ohayon and Schejter, 1999) |
| Ubq-NG-Cnn | (Wong et al, 2022) |
| Ubq-Spd-2-GFP | (Dix and Raff, 2007) |
| Spd-2^z35711 | (Giansanti et al, 2008) |
| Spd-2^G20143 | (Dix and Raff, 2007) |
| Polo-TRAP-GFP | (Buszczak et al, 2007); appears to not be fully functional and is only viable as a heterozygote. |
| ncd-γ-tubulin-37c-GFP | (Hallen et al, 2008) |
| Ubq-Msps-GFP | (Lee et al, 2001) |
| Ubq-GFP-TACC | (Barros et al, 2005) |
| tacc^stella592 | (Lee et al, 2001) |
| Ubq-Grip71-GFP | (Conduit et al, 2014b) |
| Grip75-GFP | (Tovey et al, 2021); CRISPR Knock-in fast folded GFP |
| Grip128-GFP | (Tovey et al, 2021); CRISPR Knock-in fast folded GFP |
| Ubq-Aurora A-GFP | (Lucas and Raff, 2007) |
| cycB^2 | (Jacobs et al, 1998) |
| Ubq-Spd-2-NG | Generated in this study; appears to be fully functional and rescues Spd-2^−/− mutant. |
| Ubq-Spd-2-Cdk20-NG | Generated in this study; male sterile |
| Ubq-NG-Cnn-Cdk6 | Generated in this study; male sterile |
| Ubq-Spd-2 | Generated in this study |
| Ubq-Spd-2-Cdk20A | Generated in this study; male sterile |
| Ubq-Cnn | Generated in this study |
| Ubq-Cnn-Cdk6A | Generated in this study; male sterile |
| Ubq-Spd-2-mCherry | (Alvarez Rodrigo et al, 2019) |
| Ubq-Jupiter-GFP | (Karpova et al, 2006) |
| Spd-2-NG | (Wong et al, 2022); CRISPR Knock-in Neongreen |
| PCNA-RFP | (Deneke et al, 2016) |

S1095A; S1102A; S1117A were synthesised by Genewiz and assembled with the PCR amplified backbone of pRNA-NG (CT) (Aydogan et al, 2020) using HiFi Assembler (NEB, USA) to create Spd-2-Cdk20A-NG. This was recombined to create a pDONR vector and then a destination vector using Gateway technology (Thermo Fisher Scientific). For untagged Spd-2-Cdk20A, a stop codon was reintroduced to the C-terminus of the Spd-2-Cdk20A pDONR using Q5 Site-Directed Mutagenesis (NEB, USA) and the resulting vector was recombined with a destination vector encoding no tag (Aydogan et al, 2018), using Gateway technology. To generate Cnn-Cdk6A mutants, two different cDNA fragments (from the cnn-PA isoform, most highly expressed in embryos) encoding the following amino acid substitutions S64A; S91A;

**Table 2.** *Drosophila* stocks used in specific experiments.

| Genotype | Experiments | Figure |
|---|---|---|
| Ubq-NG-Cnn, cnn$^{f04547}$/cnn$^{HK21}$ | Dynamics of total intensities or areas across nuclear cycles 11–13 | 1A,B; 3A; EV1-2A; EV4A |
| Ubq-Spd-2-GFP ; Spd-2$^{z35711}$/Spd-2$^{G20143}$ | | |
| ncd-γ-tubulin-37c-GFP | | 1A,C; 3B; EV1-2B; EV4B; S1 |
| Ubq-Msps-GFP | | |
| Ubq-GFP-TACC, tacc$^{stella592}$ | | |
| Ubq-Grip71-GFP | | 1A,D; 3C; EV1-2C; EV4C; S1 |
| Grip75-GFP | | |
| Grip128-GFP | | |
| Ubq-Aurora A-GFP | | |
| Polo-TRAP-GFP/+ | (1) Dynamics of total intensities or areas across nuclear cycles 11–13<br>(2) Polo-GFP expressed with an endogenous level of Cyclin B, which serve as a control for Cyclin B half dosage experiment | 1A,B; 3A; 8; EV1-2A; EV4A |
| Jupiter-GFP/Ubq-Spd-2-mCherry, spd-2$^{G20143}$ | Jupiter-GFP expressed with Spd-2-mCherry | 2A,B; EV3A |
| ncd-γ-tubulin-37c-GFP/+ | Tagged-centrosomal proteins expressed with an endogenous level of Cyclin B, which serve as a control for Cyclin B half dosage experiment | 4A,B; 6A,B |
| Ubq-GFP-TACC, tacc$^{stella592}$/+ | | 4C,D; 6C,D |
| Ubq-Spd-2-GFP ; Spd-2$^{z35711}$/+ | | 5E,F |
| Ubq-NG-Cnn, cnn$^{f04547}$/+ | | 5K,L |
| cycB$^2$/+ ; ncd-γ-tubulin-37c-GFP/+ | Tagged-centrosomal proteins expressed with a reduced level of Cyclin B for Cyclin B half dosage experiment | 4A,B |
| cycB$^2$/+ ; Ubq-GFP-TACC, tacc$^{stella592}$/+ | | 4C,D |
| cycB$^2$/+; Polo-TRAP-GFP/+ | | 8 |
| Ubq-NG-Cnn, cnn$^{f04547}$/cycB$^2$ | | 5K,L |
| Ubq-Spd-2-GFP/cycB$^2$ ; Spd-2$^{z35711}$/+ | | 5E,F |
| Ubq-Spd-2-NG/+ | Spd-2-NG expressed in the presence of endogenous level of wildtype Spd-2 | 5A–D |
| Ubq-Spd-2-Cdk20A-NG/+ | Spd-2-NG with 20 consensus Cdk1 phosphorylation sites mutated to alanine in the presence of endogenous level of wildtype Spd-2 | 5A–D |
| Ubq-NG-Cnn/+ | NG-Cnn expressed in the presence of endogenous level of wildtype Cnn | 5G–J |
| Ubq-NG-Cnn-Cdk6A/+ | Spd-2-NG with 6 consensus Cdk1 phosphorylation sites mutated to alanine in the presence of endogenous level of wildtype Cnn | 5G–J |
| ncd-γ-tubulin-37c-GFP/Spd-2$^{z35711}$ | γ-tubulin-37c-GFP with a reduced level of Spd-2 | 6A,B |
| cnn$^{HK21}$/+ ; ncd-γ-tubulin-37c-GFP/+ | γ-tubulin-37c-GFP with a reduced level of Cnn | 6C,D |
| Ubq-Spd-2/+; ncd-γ-tubulin-37c-GFP/+ | γ-tubulin-37c-GFP expressed with an untagged Spd-2 and an endogenous level of wildtype Spd-2 | 7A,B |
| Ubq-Spd-2-Cdk20A/+;; ncd-γ-tubulin-37c-GFP/+ | γ-tubulin-37c-GFP expressed with an untagged Spd-2-Cdk20A and an endogenous level of wildtype Spd-2 | 7A,B |
| Ubq-Cnn/+; ncd-γ-tubulin-37c-GFP/+ | γ-tubulin-37c-GFP expressed with an untagged Cnn and an endogenous level of wildtype Cnn | 7A,B |
| Ubq-Cnn-Cdk6A/+;; ncd-γ-tubulin-37c-GFP/+ | γ-tubulin-37c-GFP expressed with an untagged Cnn-Cdk6 and an endogenous level of wildtype Cnn | 7A,B |
| PCNA-RFP/ncd-γ-tubulin-37c-GFP | RCNA-RFP co-expressed with γ-tubulin-GFP | EV3 |
| WT | (1) WT flies for mRNA injection<br>(2) Untagged fly lines for immunofluorescence experiments | EV5; S3 |
| Ubq-Spd-2/+ | | |
| Ubq-Spd-2-Cdk20A/+ | | |
| Ubq-Cnn/+ | | |
| Ubq-Cnn-Cdk6A/+ | | |
| Spd-2-NG/+ | CRISPR Knock-in Spd-2-NG expressed in an endogenous level of wildtype Spd-2 | S4 |
| cycB$^2$/ + ; Spd-2-NG/ + | Spd-2-NG CRISPR Knock-in expressed with a reduced level of Cyclin B for Cyclin B half dosage experiment | |
| Ubq-Spd-2-GFP/+ ; Spd-2$^{z35711}$/Spd-2$^{G20143}$ | 1 copy of Spd-2-GFP expressed in the absence of endogenous Spd-2 | S5 |
| Ubq-Spd-2-GFP ; Spd-2$^{z35711}$/Spd-2$^{G20143}$ | 2 copies of Spd-2-GFP expressed in the absence of endogenous Spd-2 | |

S364A; S1020A; S1057; S1067 were synthesised by Genewiz and assembled as described above. The resulting construct was recombined to create a pDONR vector and then a destination vector encoding mNG (NG-Cnn-Cdk6A) using Gateway technology. For untagged Cnn-Cdk6A, the Cnn-Cdk6A pDONR (described above) was recombined with a destination vector encoding no tag. Transgenic control lines expressing NG- or untagged-WT Spd-2 or WT-Cnn were generated using the same DNA templates and methods but without mutagenesis.

## Embryo collections

Embryos were collected from plates (40% cranberry-raspberry juice, 2% sucrose, and 1.8% agar) supplemented with fresh yeast suspension. For imaging experiments, embryos were collected for 1 h at 25 °C, and aged at 25 °C for 45–60 min. Embryos were dechorionated by hand, mounted on a strip of glue on a 35-mm glass-bottom Petri dish with 14-mm microwell (MatTek), and desiccated for 1 min at 25 °C before covering with Voltalef grade H10S oil (Arkema). For RNA injection, embryos were collected for 20 min at 25 °C and immediately dechorionated by hand, mounted on a strip of glue, and desiccated for 7 min at 25 °C before covering with Voltalef. All embryos were injected with RNA within a time interval of 2 min before being aged at 25 °C for ~60 min prior to imaging.

## Immunoblotting

Immunoblotting analysis to estimate protein expression level was performed as previously described (Aydogan et al, 2018). The samples were resuspended in sample buffer (0.25 M Tris-HCl pH 6.8, 8%w/v SDS, 20% β-mercaptoethanol, 40%v/v glycerol, 0.08% Bromophenol blue) and boiled at 100 °C for 10 min on a heat block, gently spun for 5 min on a small lab bench centrifuge, and stored at −20 °C. A total of 10 µl of the sample (which is the equivalent of five embryos for Cnn and ten embryos for Spd-2) was loaded into each lane of a 3–8% Tris-Acetate pre-cast SDS-PAGE gel (Invitrogen, Thermo Fisher Scientific) and then transferred from the gel onto a nitrocellulose membrane (0.2 µm #162-0112; Bio-Rad) using a Bio-Rad Mini Trans-Blot system. For Western blotting, the membranes were incubated with blocking buffer (1× TBS, 4% milk powder, 0.1% Tween-20) for 1 h on an orbital shaker at 4 °C overnight in blocking buffer with the primary antibody (see description below). The membranes were washed 3× with TBST (TBS, 0.1% Tween-20) and then incubated for 1 h in blocking buffer with the secondary antibody (1:3000 dilution, horseradish peroxidase-conjugated for chemiluminescence analysis). The membranes were washed 3× for 15 min with TBST buffer, before incubation for 1 min in HRPO substrate (#34095, Thermo Fisher Scientific SuperSignal West Femto Maximum Sensitivity Substrate,) at a concentration that was empirically determined for each different protein and exposed to X-ray film for ~10 s to 2 min. The following primary antibodies were used: Rabbit anti-Spd-2 (1:500) [Lab stock #57] (Dix and Raff, 2007), Rabbit anti-Cnn (1:1000) [Lab stock #37] (Lucas and Raff, 2007), and Rabbit anti-GAGA factor (1:500) [Lab stock #144] (Raff et al, 1994), Mouse anti-actin (1:500) [#A3853, Sigma]. HRP-conjugated Donkey anti-Rabbit (NA934V lot:17876631, Cytiva Lifescience) and Sheep ECL anti-Mouse IgG (NA931V, GE Healthcare) secondary antibodies were used at 1:3000.

## RNA synthesis and microinjection

To generate WT Spd-2-NG construct used for RNA injection experiments, a pDONR vector containing Spd-2 CDS was recombined with a destination vector containing monomeric Neongreen CDS using Gateway technology. pRNA-Spd-2-Cdk20A-NG construct was generated as described in "Transgenic fly line generation". To generate Spd-2 Cdk20E mutants, two different cDNA fragments encoding the following amino acid substitutions: S49E; T112E; S311E; T337E; S484E; T516E; S531E, S536E; T561E; S606E; S614E; S618E; S625E; S944E; S1021E; T1023E; S1065E; S1095E; S1102E; S1117E were introduced and assembled as described in "Transgenic fly line generation". After these constructs were generated, they were digested and linearised by AscI (NEB) restriction enzyme. Linearised constructs were precipitated by 66% ethanol, 10 mM sodium acetate and 7 mM EDTA overnight at −20 °C, and dissolved in DEPC-treated $H_2O$ (Ambion). RNA was synthesised using a T3 mMESSAGE mMACHINE kit (ThermoFischer) and purified using an RNeasy MinElute Kit (Qiagen). All RNA constructs were injected into embryos at a concentration of 2 µg/µL.

## Fluorescence microscopy

Images of embryos were acquired at 23 °C using a PerkinElmer ERS spinning disk confocal system mounted on a Zeiss Axiovet 200 M microscope using Volocity software (PerkinElmer). A 63×, 1.4NA oil objective was used for all acquisition. The oil objective was covered with an immersion oil (ImmersolT 518F, Carl Zeiss) with a refractive index of 1.518 to minimise spherical aberration. The detector used was a charge-coupled device (CCD) camera (Orca ER, Hamamatsu Photonics, 15-bit), with a gain of 200 V. The system was equipped with 405 nm, 488 nm, 561 nm and 642 solid-state lasers (Oxxius S.A.). The microscope was operated using Volocity software. All red/green fluorescently tagged samples were acquired using UltraVIEW ERS "Emission Discrimination" setting. The emission filter of these images was set as follows: a green long-pass 520 nm emission filter and a red long-pass 620-nm emission filter. For dual-channel imaging, the red channel was imaged before the green channel in every slice in a z-stack. 0.5 µm z-sections were acquired, with the number of sections, time step, laser power, and exposure depending on the experiment. Imaging of fixed testes was performed using an Olympus FV1200 microscope equipped with a UPLSAPO 100XO NA:1.40 lens. The fluorescence of stained DAPI, Spd-2, Cnn and Asl were excited by a Violet Diode Laser 405 nm, Argon 488 nm, DPSS Laser 559 nm and a 635 nm Diode Laser. respectively. A "Quad" dichoric of DM405/488/559/635 nm was used. A FV10-ASW 4.2.1.20 (Olympus) software was used to operate the system.

## Data analysis

The following analysis pipeline was used (no blinding was employed, but all data were computationally analysed in the same way): Raw time-series images were corrected for photobleaching using the exponential decay function and then z-projected using the maximum intensity projection function. The background was estimated and corrected using an uneven illumination background correction (Soille, 2004). Centrosomes were tracked using the

TrackMate Plug-In (Tinevez et al, 2016) in Fiji (Schindelin et al, 2012). A custom Python script was then used to threshold and extract the fluorescence intensities and areas of all tracked centrosomes as they changed over time in each individual embryo, as previously described (Wong et al, 2022). To extract the features of Spd-2, Polo, γ-tubulin, Msps, TACC, Grip71, Grip75, Grip128 and Aurora A recruitment, we measured the *initial intensity* of the centrosomes as they first separated in early S-phase and their *maximum intensity* as their levels peaked; the time between these points represented the *growth period*, while the *growth rate* was calculated as: *(maximum intensity – initial intensity)/growth period*, and the decrease rate was calculated as: *(final intensity − maximum intensity)/the duration of the decrease*. The total intensities of Cnn, which exhibits extensive centrosomal flaring, were extracted as described previously (Wong et al, 2022). The spindle length and mean intensity were analysed in embryos co-expressing the MT marker Jupiter-GFP (Karpova et al, 2006) and Spd-2-mCherry. The spindle length was calculated by averaging the distances between all pairs of centrosomes labelled by Spd-2-mCherry. The intra-centrosomal distance was subsequently used to construct a box with an aspect ratio of (10 pixel width × spindle length–height), within which total intensities of Jupiter-GFP were averaged. The diameter and PCNA intensity were analysed in embryos co-expressing PCNA-RFP and Spd-2-GFP. An Otsu algorithm was used to extract areas and intensities of nuclei. The diameter of an individual nucleus was then calculated by this equation $\sqrt{A/\pi}$, where A is the area of a nucleus.

As mentioned in the main text, not all of the centrosome proteins could be followed for a full nuclear cycle: (1) Polo could not be followed in mitosis as it binds to the kinetochores, making it impossible to accurately track centrosomes (so the data for mitosis is missing for Polo); (2) Cnn exhibits extensive flaring at the end of mitosis/early S-phase (Megraw et al, 1999), so we could track individual separating centrosomes labelled with NG-Cnn in early S-phase until they have moved sufficiently far-apart (so the early S-phase time points are missing for Cnn).

### Analysis of spermatocytes

Testes were dissected from males of the appropriate genotype and were dissected, fixed and stained, as described previously (Roque et al, 2012). The following antibodies were used: Sheep anti-Cnn (1:500; animal SKS027, (Cottee et al, 2013)), Guinea pig anti-Asl (1:500; animal # SKC123; (Roque et al, 2012)), Rat anti-Spd-2 (1:500; animal SKR100, (Franz et al, 2013)), Alexa Fluor 488 nm-conjugated Donkey-anti-Rat IgG (1:500; A21208; Thermo Fisher Scientific), Alexa Fluor 594 nm-conjugated Donkey-anti-Sheep IgG (1:500; A11016; Thermo Fisher Scientific) and Alexa Fluor 633 nm-conjugated Pig anti-Guinea pig IgG (1:500; A21105; Thermo Fisher Scientific). Samples were mounted in Vectashield medium with DAPI (H-1200; Vector Laboratories). Meiotic cells were then scored blind for the presence/absence of cytoplasmic Spd-2 and Cnn foci.

### Statistical analysis

All data graphs were generated by GraphPad Prism 8 or 9. The details of statistical tests, sample size, and definition of the centre and dispersion are provided in individual Figure legends.

## Data availability

The processed imaging source data, extracting the centrosome fluorescent intensities from movies of embryos, is linked to each Figure. All raw imaging data has been deposited at The BioImage Archive (Hartley et al, 2022) with the accession number S-BIAD988. The custom Python script used to process the raw time-series images, extract centrosome data and parameterise growth parameters can be downloaded from the Raff lab GitHub repository via https://github.com/RaffLab/Wong-et-al-2021.

## Peer review information

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

## Acknowledgements

We thank Paul Conduit for the Grip75-GFP and Grip128-GFP CRISPR knock-in lines. We are grateful to members of the Raff Laboratory for advice, discussion and for critically reading the manuscript. The research was funded by a Wellcome Trust Senior Investigator Award (215523) to JWR (AW and SS) and a Cancer Research UK Oxford Centre Prize DPhil Studentship (C5255/A23225), a Balliol Jason Hu Scholarship, a Clarendon Scholarship, a Max Planck Croucher Postdoctoral Fellowship and a Junior Research Fellowship in Medical Sciences from the Wadham College (to S-SW). This research was funded in whole or in part by Wellcome (215523). For the purpose of Open Access, the author has applied a CC BY public copyright licence to any Author Accepted Manuscript (AAM) version arising from this submission.

## Author contributions

**Siu-Shing Wong**: Conceptualisation; Data curation; Software; Formal analysis; Investigation; Methodology; Writing—original draft; Writing—review and editing. **Alan Wainman**: Formal analysis; Investigation; Writing—review and editing. **Saroj Saurya**: Resources; Writing—review and editing. **Jordan W Raff**: Conceptualisation; Formal analysis; Funding acquisition; Writing—original draft; Project administration; Writing—review and editing.

## Disclosure and competing interests statement

The authors declare no competing interests.

# Expanded View Figures

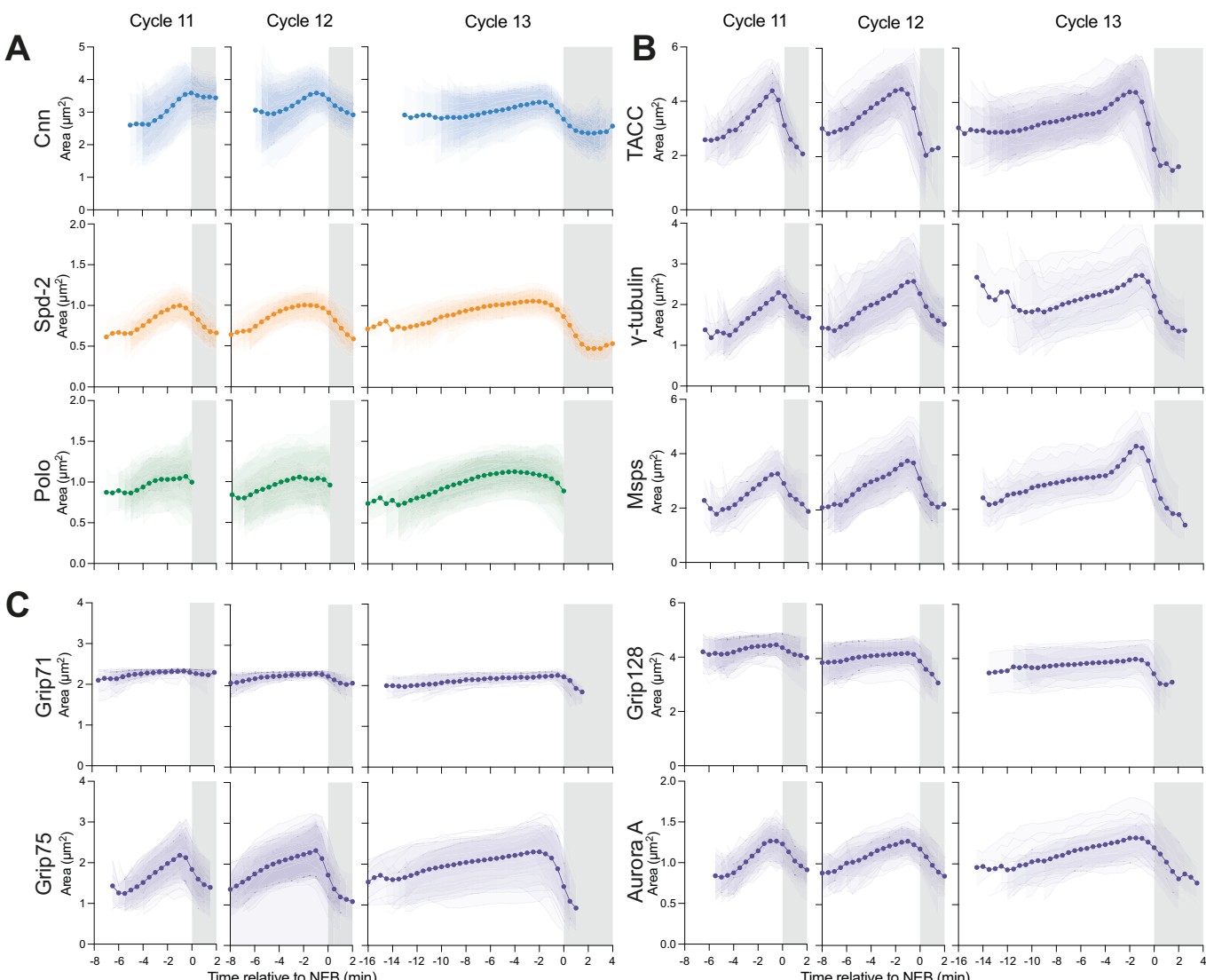

**Figure EV1.  Analysis of centrosome growth kinetics measured by changes in centrosomal area during NC11, 12 and 13.**

(A) Graphs show how the mean centrosomal area (±SD of the data in each individual embryo shown in reduced opacity) of the PCM-scaffolding proteins Cnn, Spd-2 and Polo, varies during NC11, 12, and 13. These graphs were derived from the same embryos analysed in Fig. 1A. All individual embryo tracks were aligned to NEB ($t = 0$). The white parts of the graphs represent S-phase, and the grey parts represent mitosis. (B, C) Graphs show the same as (A) for the Class I (B) and Class II (C) PCM-client proteins (graphs derived from the same embryos analysed in Fig. 1B,C). Note that the Grip71- and Grip128-fluorescent fusion proteins were very dim. As a result, although their centrosomal distribution appeared very similar to Grip75 and γ-tubulin (Fig. 1A), our computational thresholding pipeline assigned them a larger area than these other proteins. This meant that the computationally calculated area of both proteins did not change very much during each nuclear cycle. $N = 7$–15 embryos analysed at each nuclear cycle for each marker with a total of $n = $ ~200–400, ~400–800, or ~600–1200 total centrosomes analysed at NC11, 12 and 13, respectively.

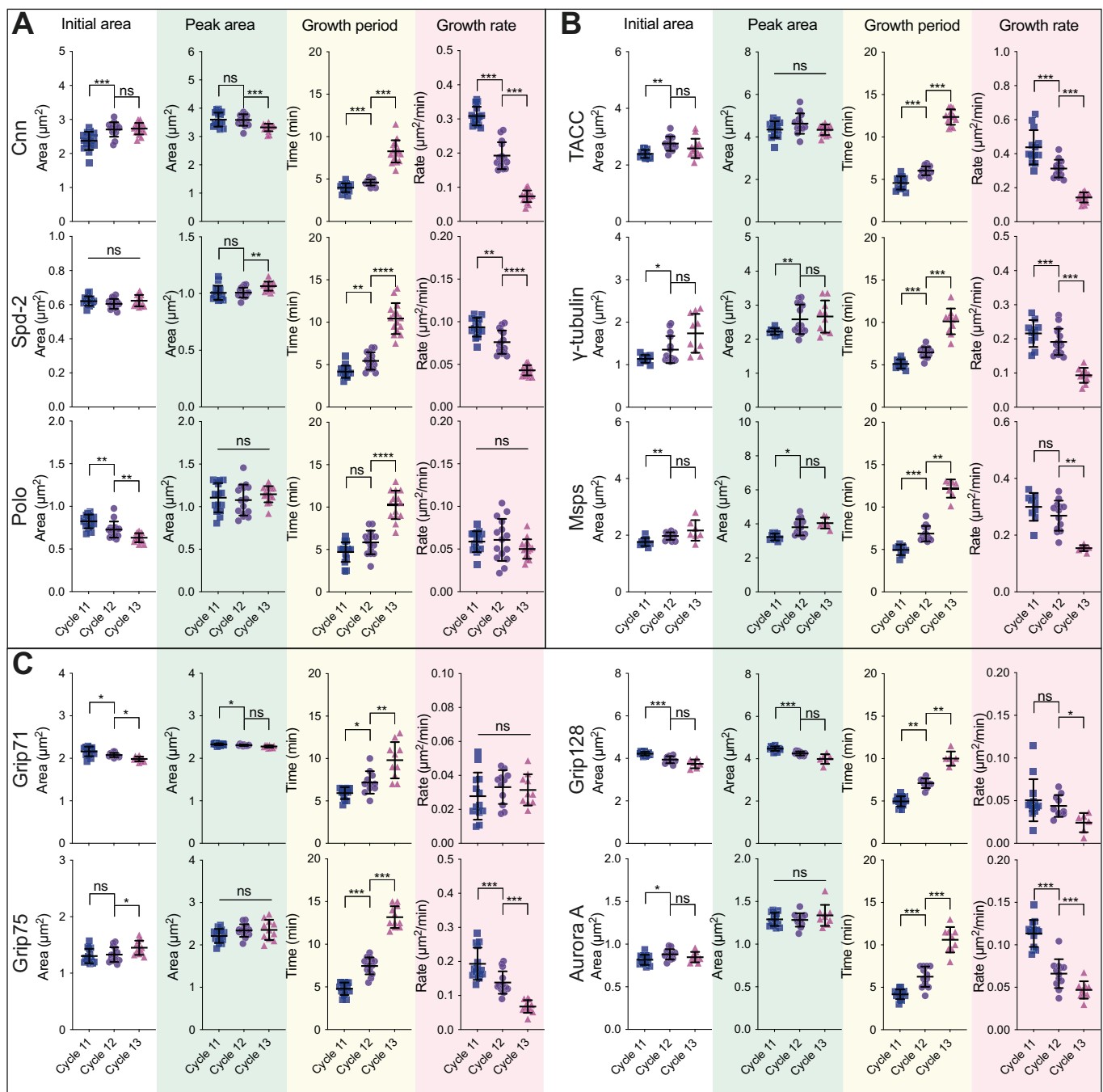

**Figure EV2. Analysis of centrosome growth parameters, measured by centrosome area, during NC11, 12 and 13.**

(A) Scatter plots show the mean (±SD) initial fluorescent intensity (left graphs), peak fluorescent intensity (boxed in green), growth period (boxed in yellow), and growth rate (boxed in pink) in NC11, 12, and 13 for the PCM-scaffolding proteins. (B, C) Scatter plots show the same as in (A) for the Class I (B) and Class II (C) PCM-client proteins. All plots were derived from the same embryos analysed in Fig. 1. Statistical comparisons were performed as described in the legend to Fig. 3.

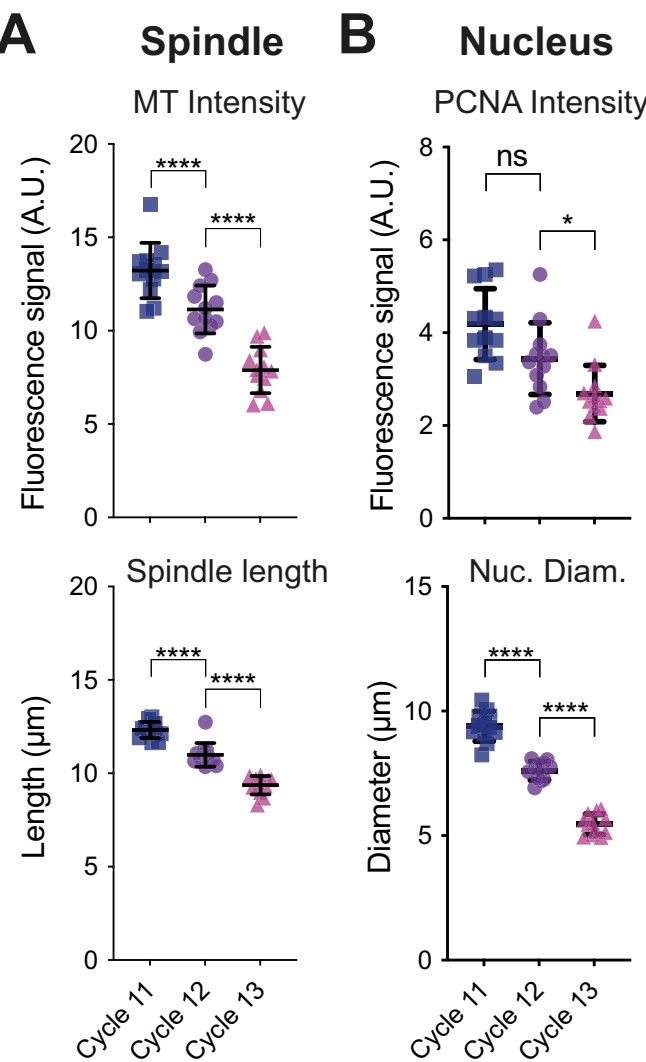

**Figure EV3.  Comparison of how maximum spindle size and maximum nuclear size change during NC11, 12 and 13.**

(A) Scatter plots compare how the mean (±SD) fluorescence intensity of the metaphase mitotic spindle (top) and the mean (±SD) metaphase spindle length (bottom) vary during NC11, 12 and 13. (B) Scatter plots compare how the mean (±SD) fluorescence intensity of nuclear PCNA-RFP (top) and the mean (±SD) nuclear diameter (bottom) vary during NC11, 12 and 13. $N = 8$–12 embryos were analysed at each nuclear cycle with a total of $n = $ ~200–300, ~400–500, or ~600–800 total spindles/nuclei analysed at NC11, 12 and 13, respectively. Statistical comparisons were performed as described in the legend to Fig. 3.

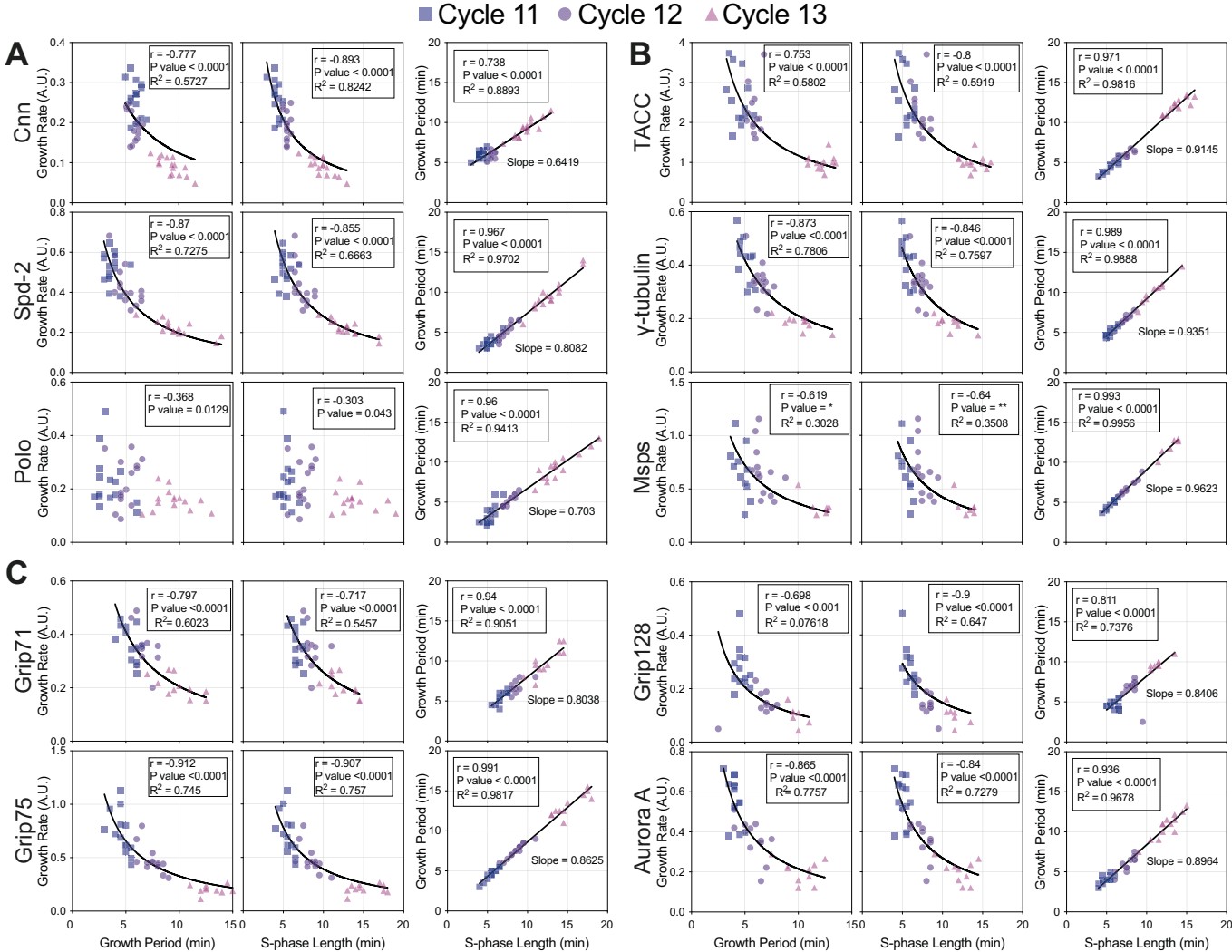

**Figure EV4. Analysis of the strength of correlation between various centrosome growth parameters during NC11, 12 and 13.**

(A) Scatter plots show the correlation between the centrosome growth rate and period (left graphs for each protein), growth rate and S-phase length (middle graphs for each protein), and growth period and S-phase length (right graphs for each protein) for the centrosome scaffold proteins. Each data point represents an individual embryo at either NC11 (deep purple squares), NC12 (light purple circles) and NC13 (pink triangles) (calculated from the data shown in Fig. 1A). Lines indicate mathematically regressed best fits for inverse (i.e., *y* is proportional to 1/*x*; left and middle graphs) and linear (i.e., *y* is proportional to *x*; right graphs) correlations. The goodness of fit ($R^2$), strength of correlation (*r*) and the statistical significance (*P* value) are indicated and were calculated in custom Python scripts and GraphPad Prism by either Pearson test (bivariate Gaussian-distributed) or Spearman test (bivariate non-Gaussian-distributed data). Bivariate Gaussian distribution was tested by Henze–Zirkler test. Note that for Polo, the correlation between the centrosome growth rate and either the centrosome growth period or S-phase length did not fit an inverse function well, although the trend was still significant (*P* < 0.05). This suggests that this relationship may be more complicated than for the other proteins, perhaps because Cdk/Cyclins and Polo influence each other's behaviour in multiple ways. (B, C) Scatter plots show the same as in (A) but for the Class I (B) of Class II (C) client proteins.

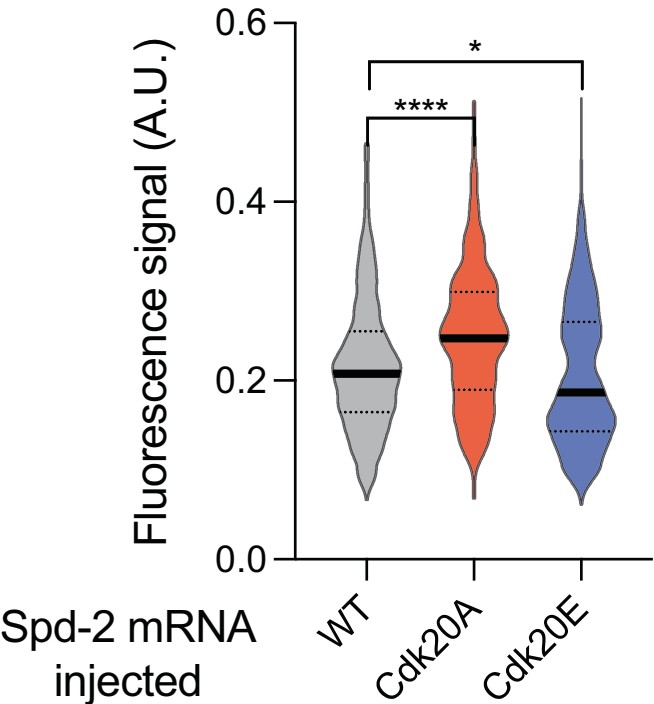

**Figure EV5.  Comparison of the centrosomal levels of WT Spd-2-NG, Spd-2-Cdk20A-NG or Spd-2-Cdk20E-NG assayed by mRNA injection.**

Violin plots quantify the centrosomal fluorescence intensity in embryos injected with mRNA encoding either WT Spd-2-NG, Spd-2-Cdk20A-NG or Spd-2-Cdk20E-NG. Horizontal bars indicate the median±quartile. Embryos were injected with mRNA (which is rapidly translated), and the fluorescence intensity of the 50 brightest centrosomes in each embryo were assayed in mid-S-phase (when Spd-2 levels are maximal) ~1 h later. $N = 12$-15 embryos, $n = 600$-750 centrosomes. Statistical significance was computed using a Kruskal–Wallis test, followed by a Dunn's multiple comparisons test (*$P < 0.05$, ****$P < 0.0001$).

