## [Peer Review File · The EMBO Journal]

Regulation of centrosome size by the cell-cycle oscillator in *Drosophila* embryos

Siu-Shing Wong, Alan Wainman, Saroj Saurya, and Jordan Raff
DOI 10.15252/embj.2023115516

Corresponding author(s): Jordan Raff (jordan.raff@path.ox.ac.uk)

Review Timeline:

Transfer from Review Commons:	5th Sep 23
Editorial Decision:	17th Oct 23
Appeal Received:	6th Nov 23
Editorial Decision:	13th Nov 23
Revision Received:	12th Dec 23
Accepted:	13th Dec 23

Editor: Hartmut Vodermaier

HfUbgUW]cb'FYdcfh 'H]g'a Ubi gW]dhik Ug'lfUbgZYffYX'lc 'H Y9A6C'>CI FB5 @Z`ck]b[
dYYf'fYj]Yk 'UhiFYj]Yk '7 ca a cbg"

Review #1

1. Evidence, reproducibility and clarity:

Evidence, reproducibility and clarity (Required)

The manuscript by Wong et al. investigates how cells regulate the increase in the size of the centrosomes (more specifically the size of the pericentriolar material or PCM) that occurs during preparation for mitosis. They use the *Drosophila* syncytial embryo as a model, focusing on nuclear cycles 11-13, during which cell cycle progression gradually slows. The authors find that centrosomes grow to a consistent size at each cycle by adjusting to the slowed cell cycle, reducing the growth rate and increasing the growth period. This adjustment is proposed to be regulated by the Cdk/Cyclin cell cycle oscillator. Curiously, Cdk/Cyclin activity seems to both promote and inhibit the increase in centrosome size, depending on whether its activity is moderate or very high, respectively. Both effects are proposed to depend on the phosphorylation of centrosome proteins by Cdk/Cyclin.

1. While being comprehensive in the number and type of markers that are being analyzed, there is no analysis of the centrosome's MTOC activity. In my opinion this is missing since centrosome size alone is not necessarily indicative of its MTOC activity, but MTOC activity is what ultimately matters for its role during mitosis. For example, it was observed that centrosome size declines already before mitotic entry, but it is possible that centrosome MTOC activity does not (similar to differences in the timing of the decline of PCM scaffold vs PCM client proteins). While not strictly related to size control, centrosome activity is biologically more relevant than solely size. I would consider it optional, if the authors decide to talk only about centrosome size, but then it should be made clear that size here may not be the most relevant factor.

2. The authors say that during NC13 PCM client proteins can be recruited in "at least two different ways" (p. 7), including a way (rapid increase before peak) that does not resemble PCM scaffold recruitment (steady increase during NC13). How can these two different ways and kinetics be determined by the same Cdk/Cyclin oscillator?

3. I am puzzled by the conclusion that Cdk/Cyclin directly phosphorylates Spd-2 or Cnn at the sites used for mutagenesis. This cannot be concluded based on the presented data.

4. Fig. 6: Doesn't the data show that Cnn does not affect the initial rate of g-tub recruitment, but only the later rapid recruitment shortly before mitosis? In contrast Spd-2 seems to affect the initial phase. This should be described more precisely. Again, I am wondering how this is compatible with direct regulation by a single

oscillator, as suggested by the authors (see also point 2 above).

5. I don't find the proposed model very convincing and not fully supported by the presented data.

First, the recruitment kinetics of different centrosome proteins are not all the same, arguing against a simple relationship based on phosphorylation by Cdk/Cyclin. For example, kinases (or phosphatases) may be recruited (or displaced) by Cdk/Cyclin at the centrosome and then locally regulate binding or maintenance of certain centrosome proteins. This could explain profiles that do not display a steady change over time, as would be expected by direct regulation by Cdk/Cyclin.

Second, it is not clear from the description in the text or from Fig. 8 how moderate Cdk/Cyclin activity can promote recruitment and high activity induce loss of proteins at centrosomes. In fact, the experiments with Spd-2 and Cnn phospho-mutants suggest that phosphorylations at the mutated sites also reduce centrosome binding during S phase (at moderate activity) and not only shortly before mitosis (at high activity), since alanine mutants of both Spd-2 and Cnn are increased at centrosomes also during S phase. The model seems to ignore this observation. If these sites are already phosphorylated to decrease centrosome binding in S phase, then what triggers the rapid decrease shortly before mitosis?

6. Can the authors identify and mutate CdK/Cyclin dependent phospho-sites in centrosome proteins that promote centrosome recruitment at moderate Cdk/Cyclin activity? As an alternative to the "protein availability" model for regulation of centrosome size, the proposed model needs to explain how a steadily increasing activity (Cdk/Cyclin) can first induce growth and then turn growth off, when the desired size is reached. This is obvious in the "protein availability" model, where the available protein steadily decreases as centrosomes grow, but this is not at all obvious for an oscillator that behaves in the opposite way during the same period and that can only phosphorylate sites that decrease centrosome binding.

****Minor:****

1. The authors observe differences in the intensity profiles for different subunits of the gamma-tubulin complex. How do they explain this? Are they not in the same complex? The authors should mention and comment on this.

2. The authors refer at various points in the manuscript to an "inverse-linear" relationship between S phase length and centrosome growth rate, but according to the graphs the rate does not change linearly.

2. Significance:

Significance (Required)

This is an interesting manuscript that reaches somewhat different conclusions regarding centrosome size control when compared to previous studies in other organisms. In particular, work in *C. elegans* has proposed that centrosome growth regulation is controlled by the limited cytoplasmic availability of PCM building blocks, whereas the current study proposes a different model based on the activity of a cell cycle oscillator. The model system and approaches are well presented and the data is of good quality. The authors monitor a large number of centrosome markers, each with detailed quantifications of intensity and distribution over time during the different cycles. They also employ two different ways of quantifying centrosome size with similar results, making their quantifications more robust. While the authors include phospho-mutants in their analyses that presumably cannot be phosphorylated by Cdk/Cyclin, the study is largely descriptive. Still, the authors present interesting observations and propose the "oscillator model" an alternative to the "limited availability model" for the regulation of centrosome size, and perhaps that of other organelles. Assuming the authors can clarify inconsistencies and/or provide additional data to support the proposed model, this could be an important finding that expands cell biologists' understanding of organellar size control.

I have expertise in centrosome biology and the role of centrosomes as MTOCs, as well as more general expertise regarding the function of the microtubule cytoskeleton in cell division and differentiation.

3. How much time do you estimate the authors will need to complete the suggested revisions:

Estimated time to Complete Revisions (Required)

(Decision Recommendation)

Between 1 and 3 months

Yes

Review #2

1. Evidence, reproducibility and clarity:

Evidence, reproducibility and clarity (Required)

Control of organelle size has been an active field of research for many years for a large variety of cellular structures and in a range of experimental models. Here, Jordan Raff and colleagues examine the mechanisms underlying centrosome (PCM) size control in *Drosophila* syncytial embryos, building on their previous work (Wong, EMBOJ 2022) to propose a role for CDK in both promoting (at intermediate levels) and inhibiting (at high levels) PCM expansion.

I found this a difficult manuscript to review, not only because the subject matter is complicated, but so is the writing. Having read and re-read the manuscript some clarity eventually emerges, but it shouldn't be that inaccessible. As for the authors' model I find it intriguing, but not fully supported by the data currently presented.

Major points

1. Central to the authors' model is the proposed dual function of Cdk (or CCO in the authors' terminology) in both promoting and inhibiting centrosomal protein accumulation. This the authors test by reducing the gene dosage of cyclin B and using putatively non-phosphorylatable versions of Spd-2 and Cnn. Both approaches to me appear quite problematic. The latter perturbation is hard to interpret given that whether these are indeed Cdk phosphosites that they have mutated is unknown and there are plenty of other possibilities how this might perturb protein function, as the authors' lack of success doing the same for gamma-tubulin illustrates. The former perturbation also lacks context. Does reducing cyclin B gene dosage reduce peak CDK activity or does it merely take longer to reach the same maximum, as appears to occur naturally as the cell cycle slows between embryonic cycles 11 and 13 (Edgar, Genes Dev 1994)? A more direct way to test their model would be to arrest the embryo in S phase (which in their model should lead to indefinite growth) or mitosis using suitable drugs/genetic perturbations. Is this not feasible in the fly system?
2. Similarly critical is that centrosomal protein accumulation is accurately measured. I am not entirely convinced that this is so. If one takes their estimations of centrosome

size at face value, then the space occupied by gamma-tubulin (slightly over 1 μm^2 peak area according to Fig. S3) is significantly smaller than that occupied by Grip128 (4 μm^2). How is this possible if these form part of the same gamma-tubulin complex? This likely reflects the fact that the dynamics of many proteins is being assessed using transgenic reporters under the control of heterologous regulatory sequences (not all of which are fully functional, eg Polo), which could result in wildly inappropriate centrosomal protein levels. It may then not be a coincidence that the centrosomal domain for Grip128 (endogenously tagged) is larger than that for gamma-tubulin (transgene).

3. Another concern is that centrosome size and integrated signal intensity do not always match, as demonstrated by Grip71 (increasing as expected during centrosome maturation in cycle 13 based on fluorescence intensity but not area - compare Figs. 1B and S1). A potential reason for this is that proteins are not uniformly distributed within centrosomes. For example, Polo and Spd2 are highly concentrated at centrioles. This impacts the ability to accurately measure protein accumulation based on 2D projections. Such inaccuracies likely will not affect estimation of when peak protein accumulation occurs, but may explain apparent differences in the kinetics of recruitment/dissociation of different components. Thus, the differences in the shape of the PCM client growth curves compared to those of Polo and Spd-2 (p6) may simply reflect the centriole concentration of the latter.

****Other points****

4. In *C. elegans* much of Polo at centrosomes is apparently inactive, particularly in the vicinity of centrioles (Cabral, Dev Cell 2019). Knowing whether this is also the case in flies would seem like important information to have, particularly when comparing signal intensities across the cell cycle.

5. Is it clear that there is less Spd2/Cnn at centrosomes in Spd-2/Cnn 1/2 gene dosage embryos, as the authors assume?

6. Would it not be relevant to also examine Polo 1/2 dosage embryos?

7. Based on the authors model, Cdk phosphorylation first drives PCM accumulation, then at higher levels inhibits. Yet, their non-phosphorylatable Spd2 mutant exhibits not only a delayed decline in centrosomal levels, but also higher initial levels (Fig. 5B). If Cdk initially promotes Spd2 activity what is their explanation for this?

****Minor/discussion points****

8. p4 "In typical somatic cells the two mitotic centrosomes need to grow to approximately the same size, as mitotic centrosome size asymmetry can lead to asymmetric spindle assembly and so to defective chromosome segregation. How centrosome growth is regulated in somatic cells is unclear, but in early *C. elegans* embryos, mitotic centrosome size appears to be set by a limiting pool of the PCM-

scaffolding protein SPD-2."

The authors here conflate absolute and relative size. Relative size matters to avoid spindle asymmetries, and centriole involvement in PCM recruitment helps to prevent this (Zwicker et al., PNAS 2014). Absolute size, which is what the authors are concerned with in this manuscript, may be important for spindle scaling, but this is not the same thing.

9. p5 "The centrosomal levels of Polo, Spd-2 and Cnn all started to increase at the start of S-phase, but whereas Cnn levels continued to rise and/or plateau as the embryos entered mitosis, the centrosomal levels of Polo and Spd-2 started to decrease before the entry into mitosis (Wong et al, 2021) (Figure 1A,B). Thus, the components of the mitotic PCM scaffold exhibit different growth kinetics, making it hard to use these proteins to define centrosome "size" at any particular point in the cell cycle." It is misleading and confusing for the reader to describe Polo and Spd2 as scaffold proteins as opposed to regulators of scaffold assembly. Presently Cnn is the only PCM protein demonstrated to have self-assembly/scaffolding properties based on the authors' own work (conduit, Dev Cell 2014; Feng, Cell 2017). There is little evidence that Polo and Spd2 form anything other than a nucleus for PCM growth.

10. p7 "The centrosomal levels of Grip71, Grip75, Grip128, and Aurora A tended to increase steadily through most of NC13, whereas TACC, Msps and γ -tubulin exhibited a noticeable increase in their recruitment rate towards the end of S-phase, shortly before their recruitment levels peaked (compare NC13 graphs in Fig. 1B). This difference was also obvious if we used centrosome area as a measure of centrosome size (Fig. S1). We conclude that PCM client proteins can be recruited to centrosomes in at least two different ways."

As discussed above apparent differences in kinetics may reflect limitations in the way protein accumulation is measured. It is hard to conceive of a reason why the Grips would display a different mode of protein accumulation from gamma-tubulin, nor is the idea of two different modes of protein accumulation picked up again later in the manuscript.

11. Since the authors mention that the duration of S phase increases between cycles 11 and 13 (p9), are there any measures for the timing of the beginning/end of S phase in each cycle?

12. One of the main findings in the landmark Woodruff paper from 2017 Cell paper was that PCM scaffold polymer could dynamically concentrate client proteins in the absence of any other factors, to an extent similar to that observed in vivo. This list did not include gamma-tubulin, which was later shown to require PLK1 phosphorylation of SPD-5 (Ohta, JCB 2021). However, it did include ZYG-9, the *C. elegans* ortholog of Msps. If client protein accumulation is an intrinsic property of the PCM scaffold, how do the authors explain that Msps departs prior to NEBD while Cnn continues to accumulate?

13. p13 "The expression of the mutant proteins did not appear to dramatically perturb

the centrosomal recruitment of γ -tubulin-GFP, except that the rate at which γ -tubulin-GFP left the centrosome as the embryos entered mitosis was reduced in both mutants compared to WT (Figure 7). This phenotype was subtle, but it was statistically significant, and it seems likely that the presence of large amounts of WT Spd-2 and Cnn in the mutant embryos (Figure 5A,F) would help to mask the potential severity of this phenotype."

This does not quite make sense. Fig. 5 shows that Spd2 dissociation is significantly slowed in the mutant condition. If Spd2 drives gamma-tubulin accumulation (as Fig 6 shows), then the continued presence of Spd2 should prevent dissociation. Yet it apparently does not. Why?

****Other****

14. p3 and following. The reference for the authors' prior work on PCM recruitment (Wong et al, 2021) should probably be for the final, published article in EMBO J, not the 2021 preprint.

15. Fig. 1. legend "Note that for technical reasons not all of the centrosome proteins could be followed for the full time period." Why not?

16. Figs 4-6. Which cycle is being assessed here?

17. Fig 6. Not plotted here is the rate of dissociation of gamma-tubulin, unlike eg in Fig 7. It is notable that both accumulation and dissociation appear to be slowed in the Spd2 1/2 gene dosage condition.

18. Fig S1B. Some of the graphs in this figure are not labeled (based on Fig.1 presumably gamma-tubulin and Msps).

19. Some of the data in the main figures, including the entirety of Figs. 2 and 3, could be moved to Supplemental to present a more crisp and accessible manuscript.

20. While I sympathize with the authors needing to repeat entire sets of experiments I am not entirely sure it is appropriate to recycle entire sets of data from a previous publication of theirs (Cnn, Spd-2 and Polo recruitment kinetics, reproduced from Wong et al., EMBOJ 2022), since this manuscript is largely concerned with apparent differences between the kinetics of those components and the PCM client proteins now being analysed.

2. Significance:

Significance (Required)

Control of organelle size has been an active field of research for many years for a large variety of cellular structures and in a range of experimental models. Here, Jordan Raff and colleagues examine the mechanisms underlying centrosome (PCM) size control in *Drosophila* syncytial embryos, building on their previous work (Wong,

EMBOJ 2022) to propose a role for CDK in both promoting (at intermediate levels) and inhibiting (at high levels) PCM expansion.

I found this a difficult manuscript to review, not only because the subject matter is complicated, but so is the writing. Having read and re-read the manuscript some clarity eventually emerges, but it shouldn't be that inaccessible. As for the authors' model I find it intriguing, but not fully supported by the data currently presented.

3. How much time do you estimate the authors will need to complete the suggested revisions:

Estimated time to Complete Revisions (Required)

(Decision Recommendation)

Between 3 and 6 months

No

Review #3

1. Evidence, reproducibility and clarity:

Evidence, reproducibility and clarity (Required)

In this manuscript, the authors investigated growth control of PCM at the mitotic centrosomes in late stages of the *Drosophila* syncytial embryos. They observed that

mitotic centrosomes reach to the correct sizes through 13 rounds of nuclear division by reciprocally slowing their growth rate and increasing their growth period. They assumed that the Cdk/Cyclin cell cycle oscillator (CCO) is a main controller, based on their previous works (Aydogan et al., 2018, 2020; 2022). They determined the recruitment dynamics of the key mitotic PCM scaffolding proteins (Spd-2, Polo and Cnn) and PCM-client proteins (γ -tubulin, Msps, TACC, GFP, Grip75, Grip128 and Aurora A) in living embryos, and proposed that moderate levels of the CCO activity promote centrosome growth by stimulating Polo recruitment to centrosomes, while higher levels of activity subsequently inhibit centrosome growth by phosphorylating centrosome proteins, such as Spd-2, to decrease their centrosome recruitment and/or maintenance as the embryos enter mitosis.

Experiments were cleverly designed and carefully executed. The results are nicely presented, the manuscript is clearly written, and their proposal draws a strong attention. However, in order to publish the manuscript in a prestigious journal, the authors may provide additional experimental evidence to support their proposal.

- It is very significant that the centrosome levels of Spd-2-Cdk20A-NG is stronger than Spd-2-NG throughout the cell cycle (Figure 5B,C). However, this is only an experimental evidence to support that Cdk/Cyclins directly phosphorylate Spd-2 in the run-up to mitosis to help reduce Spd-2's centrosome recruitment and/or maintenance. As the authors confessed, recruitment of Spd-2-NG to the centrosomes in CycB1/2 embryos (Figure 5D,E) may be moderate or not significant at least in this reviewer's eyes. It is worth to perform the same experiments with a phospho-mimetic Spd2-Cdk20E-NG mutant.
- The authors proposed that moderate levels of CCO activity promote centrosome growth by stimulating Polo recruitment to centrosomes. They provided an indirect evidence that centrosome levels of polo were strongly reduced in CycB1/2 embryos (Figure 4E,F). It is worth to determine the centrosome levels of Spd-2 in the Polo1/2 embryos and/or the centrosome levels of Polo phospho-resistant Spd-2 (Spd-2-Polo#A-NG).
- TACC may be an ideal PCM-client protein, apart from its importance in spindle formation in comparison to γ -tubulin (Figure 4C,D). Therefore, it is worth to perform the Figure 7 experiments with TACC.

2. Significance:

Significance (Required)

Experiments were cleverly designed and carefully executed. The results are nicely presented, the manuscript is clearly written, and their proposal draws a strong

attention. However, in order to publish the manuscript in a prestigious journal, the authors may provide additional experimental evidence to support their proposal.

3. How much time do you estimate the authors will need to complete the suggested revisions:

Estimated time to Complete Revisions (Required)

(Decision Recommendation)

Between 3 and 6 months

Yes

Full Revision

A Ubi gW]dhibi a VYf. #RC-2023-01980
7 cffYgdc bX]b[`U h cffgk Jordan, Raff

[Please use this template only if the submitted manuscript should be considered by the affiliate journal as a full revision in response to the points raised by the reviewers.

*If you wish to submit a preliminary revision with a revision plan, please use our "Revision Plan" template. **It is important to use the appropriate template to clearly inform the editors of your intentions.**]*

1. General Statements [optional]

We thank the Reviewers for their helpful and constructive comments. In response to these suggestions we have performed new experiments and amended the manuscript, as we describe in our detailed response below.

FYj Yk Yf. %

1. The Reviewer notes that while our analysis of centrosome size was comprehensive, we provided no analysis of centrosomal MTs, pointing out that while centrosome size declines as the embryos enter mitosis, the ability of centrosomes to organise MTs might not. This is a good point, and we now provide an analysis of centrosomal-MT behaviour (Figure 2). We find that there is a dramatic decline in centrosomal MT fluorescence at NEB, although the pattern of centrosomal MT recruitment prior to NEB is surprisingly complex.

2. The Reviewer questions how PCM client proteins can be recruited in different ways by the same Cdk/Cyclin oscillator. We apologise for not explaining this properly. It is widely accepted that Cdk/Cyclins drive cell cycle progression, in part, by phosphorylating different substrates at different activity thresholds (e.g. Coudreuse and Nurse, *Nature*, 2010; Swaffer et al., *Cell*, 2016). Moreover, it is also clear that Cdk/Cyclins can phosphorylate the *same* protein at different sites at different activity thresholds (e.g. Koivomagi et al., *Nature*, 2011; Asafa et al., *Curr. Biol.*, 2022; Ord et al., *Nat. Struct. Mol. Biol.*, 2019). Thus, we hypothesise that rising Cdk/Cyclin cell cycle oscillator (CCO) activity phosphorylates multiple proteins at different times and/or at different sites to generate the complicated kinetics of centrosome growth. We now explain this point more clearly throughout the manuscript.

3. The Reviewer is puzzled as to how we conclude that Cdk/Cyclins phosphorylate Spd-2 and Cnn at all the potential Cdk/Cyclin phosphorylation sites we mutate in our study. The

Reviewer is right that we cannot make this conclusion, and we did not intend to make this claim. As we now clarify (p11, para.1), although it is unclear if Cdk/Cyclins phosphorylate Spd-2 or Cnn on all, some, or none of these sites, if either protein *can* be phosphorylated by Cdk/Cyclins, then these mutants should not be able to be phosphorylated in this way—allowing us to address the potential significance of any such phosphorylation. We now also note that several of these sites have been shown to be phosphorylated in embryos in Mass Spectroscopy screens (Figure S6).

4. The Reviewer highlights differences in how Spd-2 and Cnn help recruit γ -tubulin to centrosomes (Figure 6). They ask for a more detailed description, and are puzzled as to how this is compatible with direct regulation by a single oscillator. We now explain our thinking on this important point in much more detail. It appears that Spd-2 helps recruit γ -tubulin throughout S-phase, while Cnn has a more prominent role in late S-phase (Figure 6). This is consistent with our overall hypothesis of CCO regulation, as we postulate that low-level CCO activity promotes the Spd-2/ γ -tubulin interaction in early S-phase, while higher CCO activity promotes the Cnn/ γ -tubulin interaction in late-S-phase, potentially explaining the increase in the rate of γ -tubulin (but not γ -TuRC) recruitment we observe at this point (see minor comment #1, below, for an explanation of the various γ -tubulin complexes in flies). This is consistent with recent literature showing that CCO activity promotes γ -tubulin (but not γ -TuRC) recruitment by Cnn/SPD-5 in worms and flies (Ohta et al., 2021; Tovey et al., 2021).

5. The Reviewer was not convinced by our model (Figure 8, now Figure 9), raising two major concerns. First, they were unsure how a single oscillator could generate different patterns of protein recruitment. We addressed this in point #2 and #4, above, where we explain how different *thresholds* of CCO activity trigger different events, so there is no expectation that we should observe steady changes in recruitment over time as CCO activity rises. Second, they questioned how modest levels of Cdk/Cyclin activity can promote recruitment, while high levels of activity can inhibit recruitment. In point #1, above, we cite several examples where such positive and negative regulation by different Cdk/Cyclin activity levels have been described. We also now explain throughout the manuscript why this hypothesis provides a plausible explanation for our results: with moderate CCO activity promoting Spd-2-dependent PCM-client recruitment in early S-phase; higher CCO activity promoting a decrease in Spd-2 recruitment in mid-late-S-phase (so centrosomal Spd-2 levels decline); and even higher levels of CCO activity leading to a decrease in the interactions between the client proteins and the Spd-2/Cnn scaffold as the embryos enter mitosis (so the client proteins are rapidly released from the centrosome).

The Reviewer also raised the important point here that our model does not explain why the mutant forms of Spd-2 and Cnn accumulate to higher levels at the start of S-phase, and not just at the end of S-phase/entry into mitosis. We apologise for not explaining this properly. The accumulation of the mutant proteins (particularly Spd-2, Figure 5C) in *early*-S-phase occurs because the excess mutant protein that accumulates at centrosomes in *late*-S-phase/mitosis is not removed properly from centrosomes during mitosis (presumably because there is insufficient time). Thus, centrosomes still have too much mutant Spd-2 at the start of the next S-phase. We

show this in **Reviewer Figure 1** (attached to this letter), which tracks Spd-2 behaviour further into mitosis, and now explain this in more detail in the text (p12, para.1).

7. The Reviewer questions how the CCO can both induce centrosome growth and also switch it off, as it is unclear how an oscillator that only phosphorylates sites to decrease centrosome binding could also promote growth. They ask if we can identify and mutate any Cdk/Cyclin sites in centrosome proteins that promote centrosome recruitment. As we now clarify, we did not intend to claim that the CCO *only* phosphorylates sites that decrease the centrosome binding of proteins, although we do hypothesise that such phosphorylation is important for switching off centrosome growth in mitosis. In addition, we hypothesise that moderate levels of CCO initially promote centrosome growth, and our data suggests that the CCO does this, at least in part, by promoting Polo recruitment (Figure 8). We speculate that the CCO phosphorylates specific Polo-box-binding sites in Ana1 and Spd-2, the main proteins that recruit Polo to centrioles. We agree that identifying these sites is an important next step, but it is complicated as our studies indicate that multiple sites contribute in a complex manner. Importantly, it is well established that the CCO triggers centrosome growth as cells prepare to enter mitosis, so our hypothesis that moderate levels of CCO activity initiate centrosome growth is not new or controversial.

Reviewer 7 comments

1. The reviewer asks how we explain the different incorporation profiles we observe for the different subunits of the γ -tubulin ring complex. We apologise for not discussing this point. In flies there is a “core” γ -tubulin-small complex (γ -TuSC) and a larger γ -tubulin-ring complex (γ -TuRC) that contains the Grip71, Grip75 and Grip128 subunits we analyse here (Oegema et al., JCB, 1999). The γ -TuSC functions independently of the γ -TuRC so γ -tubulin and γ -TuRC components can behave differently.

2. The Reviewer questions why we claim an “inverse-linear” relationship between S-phase length and the centrosome growth rate when the relationship is not linear (Figure 3, now Figure S3). I was originally confused by this as well but, mathematically, a linear relationship means y is proportional to x , whereas an inverse-linear relationship means y is proportional to $1/x$. Thus, an inverse-linear relationship between x and y does not plot as a straight line, but rather as the curves we show on the graphs. We now explain this in text (p9, para.2).

Reviewer 8 comments

This Reviewer found the manuscript hard to follow, so we are very grateful that they took the time to try to understand it. We agree that the subject matter is complicated, and that our presentation was not always helpful. The Reviewer’s comments have been very useful in helping us to identify (and hopefully improve) areas of particular difficulty.

Author's reply:

1. The Reviewer highlights that the two experimental approaches underpinning our main conclusions are problematic: (1) Experiments with mutants of Spd-2 and Cnn that theoretically cannot be phosphorylated by Cdk/Cyclins are hard to interpret as these mutations may have other effects; (2) It is unclear whether reducing Cyclin B levels reduces peak CDK activity or simply slows the time it takes to reach peak levels. They suggest a more direct test of our model would be to analyse PCM recruitment in embryos arrested in S-phase or mitosis.

(1) We agree that the mutations designed to prevent Cdk/Cyclin phosphorylation could perturb function in other ways, but this is true for any such mutation, and there are many papers that infer a function for Cdk/Cyclin phosphorylation from such experiments. Importantly, the centrosomal accumulation of the phospho-null mutants actually slightly increases compared to WT (Figure 5C and I), and we now show that the centrosomal accumulation of a phosphomimicking Spd-2-Cdk20E mutant slightly decreases (Figure S8). We now acknowledge the potential caveat of a non-specific perturbation of protein function, but feel that the reciprocal behaviour of the phospho-null and phospho-mimicking mutants somewhat mitigates this concern (p12, para.2). (2) Fortunately, and as we now clarify, it has recently been shown that reducing Cyclin levels does not reduce peak Cdk activity, but rather slows the time it takes to reach peak activity (Figure 2A, Hayden et al., *Curr. Biol.*, 2022). Thus, the cyclin half-dose experiments provide an excellent alternative test of our hypothesis as they show that the WT proteins can exhibit similar behaviour to the mutants if the rate of Cdk/Cyclin activation is slowed. We feel the evidence supporting our hypothesis is strong enough that it warrants serious consideration.

The suggestion to look at PCM recruitment in embryos arrested in either S-phase or M-phase is a good one, but these experiments produce complicated data. In M-phase arrested embryos, for example, Cnn levels continue to rise (see Figure 1G, Conduit et al., *Dev. Cell*, 2014), but the other PCM proteins do not (unpublished); in S-phase arrested embryos (arrested by mitotic cyclin depletion) centrosomes continue to duplicate, but now do so asynchronously, greatly complicating the analysis (McClelland and O'Farrell, *Curr. Biol.*, 2008; Aydogan et al., *Cell*, 2020). The centrosomes that don't duplicate, however, reach a constant steady-state size (where the rate of centrosome protein addition is balanced by the rate of loss). These observations are consistent with our recent mathematical modelling of mitotic PCM assembly (Wong et al., 2022) if we additionally account for cell cycle regulation (which was not considered in our original model). We believe such analyses are beyond the scope of the current paper and we plan to publish a second paper incorporating our new hypothesis into our mathematical modelling.

2. The Reviewer questions whether our methods accurately measure centrosomal protein accumulation, pointing out that γ -tubulin and Grip128 occupy different centrosomal areas—which should not be possible if they are part of the same complex. They suspect that our use of different transgenes with different promoters could explain these differences. As we should have described (see point #1 in our response to the minor comments of Reviewer #1), γ -tubulin exists in two complexes in flies, only one of which contains Grip128, so γ -tubulin and Grip128 exhibit different localisations. Moreover, as we now show (Figure S2), using

different promoters does not seem to make a difference to overall recruitment kinetics. Thus, we are confident that our methods measure centrosome protein recruitment dynamics accurately.

3. The Reviewer is concerned that our measurements of centrosome size based on fluorescence intensity (Figure 1) and centrosomal area (Figure S1) do not always match. They suggest a potential reason for this is that proteins are not uniformly distributed within centrosomes, and this may impact our ability to measure protein accumulation based on 2D projections (noting, for example, that Polo and Spd-2 are concentrated at centrioles and in the PCM, potentially explaining the different shape of their growth curves compared to the client proteins). When the centrosome-fluorescence-intensity and centrosome-area recruitment profiles of a protein do not match, the average “centrosome-density” of that protein must be changing over time. In some cases, we understand why density changes. Cnn, for example, stops flaring outwards on the centrosomal MTs during mitosis so its centrosomal area decreases even as its fluorescence intensity increases (leading to an increase in its centrosomal-density). We agree (and now discuss—p19, para.3) that the prominent accumulation of Spd-2 and Polo at centrioles could help to explain why Spd-2 and Polo accumulation dynamics differ from the client proteins.

Other points:

4. The Reviewer suggests it would be good to know how much Polo at the centrosome is active. We agree, but although commercial antibodies against PLK1 phosphorylated in its activation loop work in cultured fly cells, we cannot get them to work in embryos. Moreover, the recruitment of Polo/PLK1 to its site of action by its Polo-Box Domain is sufficient to partially activate the kinase independently of phosphorylation (Xu et al., *NSMB*, 2013). Thus, it seems likely that all the Polo/PLK1 recruited to centrosomes will be at least partially activated, even if it is not necessarily phosphorylated on its activation loop.

5. The Reviewer asks if it is clear that less Spd-2 and Cnn are recruited to centrosomes in the half gene-dosage embryos. We apologise for not mentioning that this is indeed the case. We showed this previously for Cnn (Conduit et al., *Curr. Biol.*, 2010) and we now state that this is also the case for Spd-2. We do not show the Spd-2 data as we plan to publish a comprehensive dose-response curve of Spd-2 (and Cnn) recruitment in our next modelling paper.

6. Would it not be relevant to examine Polo ½ dosage embryos? We do have this data (**Reviewer Figure 2**), attached to this letter, but it is quite complicated to interpret (as we explain in the legend). We feel it would be more appropriate to include this in our next modelling paper where we can properly explain the behaviours we observe. Publishing this data here would distract from our main message without changing any of our conclusions.

7. The Reviewer asks why the non-phosphorylatable Spd-2 protein is also present at higher levels on centrosomes at the start of S-phase (not just the end of S-phase). This was also raised by Reviewer #1 (point #5), so please see the second paragraph of our response there.

Minor/Discussion Points:

8. We thank the Reviewer for highlighting that absolute and relative centrosome size control are different things and we have amended the manuscript accordingly.

9. The Reviewer questions whether it is accurate to describe Spd-2 and Polo as scaffold proteins, noting that only Cnn has been shown to have scaffolding properties. There is strong evidence that Spd-2 has Cnn-independent scaffolding properties in flies (e.g. Conduit et al., *eLife*, 2014), but this is a fair point for Polo. We think it is justified to separate Polo from other client proteins as Polo is essential for scaffold assembly, whereas other client proteins are not. We now define our scaffold/client terminology to avoid confusion (p4, para.3).

10. The Reviewer highlights several points related to differences in recruitment kinetics (also touched on in points #2 and #3, above), noting we don't discuss properly the idea of two different modes of PCM recruitment. These are all good points, largely addressed in our response to points #2 and #3, above. We now discuss much more prominently the two different modes of client protein recruitment throughout the manuscript.

11. As we now clarify, in all our experiments we use centrosome separation and nuclear envelope breakdown (NEB) to define the start and end of S-phase, respectively.

12. The Reviewer quotes the landmark Woodruff paper (Cell, 2017) as showing that the ability to concentrate client proteins (including ZYG-9, the worm homologue of Msps) is an intrinsic property of the PCM scaffold, so how do we explain that Msps departs prior to NEB while Cnn continues to accumulate? It is indeed a striking observation of our study that all PCM client proteins (not just Msps) start to leave the centrosome prior to NEB, even as Cnn levels continue to accumulate. Our hypothesis is that this 'leaving' event is triggered by a threshold level of Cdk/Cyclin activity—explaining why these client proteins all start to leave the PCM at the same time (just prior to NEB) irrespective of nuclear cycle length. This is not incompatible with the Woodruff paper, which did not attempt to reconstitute any potential regulation by Cdk/Cyclins in their in vitro studies.

13. The Reviewer questions why Spd-2 that cannot be phosphorylated by Cdk/Cyclins (Spd-2-Cdk20A) accumulates abnormally at centrosomes in late S-phase, yet γ -tubulin (which is recruited by Spd-2) seems to leave centrosomes more slowly in the presence of the mutant protein. As we now explain more clearly, there is no contradiction here. Spd-2-Cdk20A accumulates to abnormally high levels in late-S-phase/early mitosis (Figure 5C), and this reduces the γ -tubulin dissociation rate, as we would predict (Figure 7B, right most graph). It does not "prevent" dissociation, however, (as the Reviewer seems to suggest it should?), but this is probably because these experiments have to be performed in the presence of large amounts of the WT Spd-2 (Figure 5A).

14. The referencing error has been corrected.

15. The Reviewer asks why in Figure 1 not all of the centrosome proteins could be followed for the full time period (as we mention in the legend, but do not explain). There are different reasons for different proteins: (1) Polo cannot be followed in mitosis as it binds to the kinetochores, making it impossible to accurately track centrosomes (so the data for mitosis is missing for Polo); (2) Cnn exhibits extensive flaring at the end of mitosis/early S-phase (Megraw et al., JCS, 1999), so we cannot track individual separating centrosomes labelled with NG-Cnn in early S-phase until they have moved sufficiently far-apart (so the early S-phase time-points are missing for Cnn); (3) In addition, several of the client proteins bind to the mitotic spindle, so although we can still track and measure the centrosomes in late mitosis in the graphs, we don't show pictures of these late mitosis centrosomes in the montage in Figure 1A as the images look a bit odd. We now explain these reasons in the Materials and Methods.

16. We now indicate that nuclear cycle 12 (NC12) is being analysed in Figures 4-8.

17. The reviewer questions why we don't show the decrease rate for γ -tubulin in Figure 6 (the Spd-2 and Cnn half-dose experiments), when we do show it in Figure 7 (the Spd-2 and Cnn Cdk-mutant experiments), suspecting that it is slowed in both cases. The reviewer is correct and we now show this data for both sets of experiments.

18. We have corrected the labelling error in Figure S1.

19. The Reviewer suggest moving some of the data from the main Figures, and the entirety of Figures 2 and 3 to the Supplemental Information. We understand this point, and agree that the amount of data presented in Figures 1-3 is somewhat overwhelming. We have played around with the Figures a lot—in particular trying to show a few examples of the data and moving the rest to Supplementary—but it is hard to pick a “typical” example, and the power of comparing the behaviour of so many different centrosome proteins is somewhat lost. We have tidied up several Figures and, as a compromise, we keep Figure 2 (now Figure 3) in the main text, but have moved Figure 3 to Supplementary (now Figure S5).

20. The Reviewer suggests that we should repeat the analysis of Spd-2, Polo and Cnn dynamics that we show here, as we already presented this data in a previous publication (Wong et al., EMBO. J, 2022). We understand this point, but feel this would be a less accurate comparison, as essentially all of the data shown in Figure 1 was obtained several years ago during a contiguous ~6month period. Since then, the lasers and software on our microscope system have been updated, so it would probably be less fair of a comparison to obtain new data for a subset of these proteins (and it seems overkill to perform the entire analysis again). We clearly state that this data has been presented previously, so we hope the Reviewer will agree that it is acceptable to present it again here so readers can more easily compare the data.

Reviewer #3

This Reviewer is broadly supportive of the manuscript, but to publish in a prestigious journal they think additional experimental evidence will be required to support our hypothesis.

The Reviewer notes that our only evidence that Cdk/Cyclins directly phosphorylate Spd-2 comes from our analysis of the Spd-2-Cdk20A mutant, as the effect of reducing Cyclin B dosage on WT Spd-2 behaviour is very modest. They request that we analyse the behaviour of a Spd-2-Cdk20E phospho-mimicking mutant. The effect of halving the dose of Cyclin B on Spd-2 behaviour is modest, but this is what we would predict as all we are doing in this experiment is slowing S-phase by ~15%, so Spd-2 should accumulate at centrosomes for a slightly longer time and to a slightly higher level (as we observe, Figure 5E). A great advantage of the early fly embryo system is that we can compare the behaviour of many hundreds of centrosomes, so even subtle differences like this are usually meaningful. To illustrate this point, we have now repeated the Spd-2 analysis in WT and *CycB*^{1/2} embryos (but now using a CRISPR/Cas9 Spd-2-NG knock-in line) and we see the same subtle differences (Figure S9). In addition, as requested, we have now analysed the behaviour of a Spd-2Cdk20E mutant protein using an mRNA injection assay (as it would have taken too long to generate and test new transgenic lines). In this assay we injected embryos with mRNA encoding either WT Spd-2-GFP, Spd-2-Cdk20A-GFP or Spd-2-Cdk20E-GFP. The mRNA is quickly translated, and we computationally measured the fluorescence intensity of the centrosomes in mid-S-phase (i.e. at the Spd-2 peak) (Figure S8). This analysis confirms that Cdk20A accumulates to slightly higher levels, and reveals that Cdk20E accumulates to slightly lower levels, than the WT protein. Together, these new experiments strongly support our original conclusions.

The Reviewer notes that we propose that the CCO initially promotes centrosome growth by stimulating Polo recruitment to centrosomes, but states that we only provide indirect evidence for this by showing that centrosomal Polo levels are strongly reduced in Cyclin B half-dose embryos. They suggest we determine Spd-2 levels in Polo half-dose embryos, and/or the centrosome levels of mutant forms of Spd-2 that cannot be phosphorylated by Polo. We believe the Cyclin B half-dose experiment provide *direct* support for our hypothesis that Cdk/Cyclin activity influences Polo recruitment (Figure 8), although, clearly, we have not identified the mechanism. We do, however, suggest a plausible mechanism: Ana1 and Spd-2 are largely responsible for recruiting Polo to centrosomes, and we have previously shown that several of the potential phosphorylation sites in these proteins that help recruit Polo to centrosomes are Cdk/Cyclin or Polo phosphorylation sites (Alvarez-Rodrigo et al., *eLife*, 2020 and *JCS*, 2021; Wong et al., *EMBO J.*, 2022). We are currently testing this hypothesis, but progress is slow as it is clear that multiple sites in both proteins can influence this process.

As the Reviewer requests, we have now also examined how Spd-2 and Cnn behave in Polo half-dose embryos (**Reviewer Figure 2**, attached to this letter). As we describe in the Figure legend, this data is informative, but is complicated. With relatively minor, but mechanistically important, tweaks to our previous mathematical modelling we can explain these behaviours, but introducing

Full Revision

such a significant mathematical modelling element would be beyond the scope of this paper. As described above, these findings will form the basis of a follow-up paper that is more mathematically oriented.

It is a great idea to look at mutant forms of Spd-2 that cannot be phosphorylated by Polo, but the consensus Polo phosphorylation site (N/D/E-X-S, with the N/D/E at -2 and the S at 0 being preferences, rather than a strict rule) is less well-defined than the consensus Cdk/Cyclin phosphorylation site (where the Pro at -1 is essentially invariant). Thus, we cannot accurately predict which sites would need to be mutated to generate such a mutant.

The Reviewer requests that we analyse the behaviour of TACC in embryos expressing the Spd-2-Cdk20A and Cnn-Cdk6A (as we do in Figure 7 for γ -tubulin). This is a reasonable request, but we prefer not to show this data as we have recently identified an interesting interaction between TACC, Spd-2 and Aurora A that will be the subject of another paper we hope to submit shortly. This data is hard to interpret without explaining these interactions properly, which is beyond the scope of the current manuscript.

We hope the Reviewers will agree that these changes have improved the manuscript substantially, and that it is now suitable for publication. We would like to thank them again for taking the time to read this rather complicated paper so thoroughly.

Yours sincerely,

Reviewer Figure 1

Analysis of Spd-2 and Spd-2-Cdk20A behaviour during mitosis.

Graph compares the behaviour of WT-Spd-2-NG and Spd-2-Cdk20A-NG during mitosis of NC12. The mutant protein accumulates on centrosomes to higher levels than normal during late S-phase (see Figure 5C,D); as a result, it is present at higher levels when the embryos enter mitosis, and this higher level is maintained throughout mitosis (as shown here). This means that when the embryos enter the next nuclear cycle, the centrosomes already have elevated levels of Spd-2-Cdk20A. This explains why the centrosomal levels of this protein are elevated even at the start of each S-phase, rather than just towards the end of each S-phase.

Reviewer Figure 2

Mathematical Model

Experimental Data

Analysis of Cnn and Spd-2 behaviour in *Polo*^{1/2} embryos.

(A,B,D,E) Graphs show mathematical predictions (based on the model of PCM scaffold assembly proposed in Wong et al., *EMBO J.*, 2022) (A,D) or experimental measurements (B,E) comparing Cnn (A,B) or Spd-2 (D,E) behaviour in WT and *Polo*^{1/2} embryos. (C,F) Scatter plots compare various cell cycle and centrosome growth parameters in either WT (blue or orange) or *Polo*^{1/2} embryos (green). Statistical significance was assessed using an unpaired t-test (**: P<0.01, ***: P<0.001, ns: not significant).

Cnn behaves largely as predicted by our previous mathematical modelling, and its centrosomal levels are significantly reduced in the *Polo*^{1/2} embryos. The measured difference in Cnn recruitment, however, is not as large as that predicted by the modelling, and this seems to be because Cnn continues to accumulate at centrosomes for longer than normal in *Polo*^{1/2} embryos because the cell cycle is considerably slowed in these embryos, which is not accounted for in our original model. Moreover, because Polo is intimately involved in both cell cycle progression and Cnn scaffold assembly, we cannot be sure whether the slow growth of the Cnn scaffold in *Polo*^{1/2} embryos is due to slowed scaffold assembly, slowed cycle progression, or both.

Intriguingly, Spd-2 does not behave in *Polo*^{1/2} embryos as predicted in our original mathematical modelling, as, to our great surprise, it's centrosomal levels are hardly perturbed (although its growth period is extended, as S-phase length is extended). We have realised that by simply adjusting our model so that Spd-2 recruitment to the *centriole* (but not to the *PCM*) exhibits zero-order kinetics we can explain the observed behaviour. This is biologically justifiable, and has some important implications for scaffold assembly that we are currently testing.

In summary, these experiments are interesting and potentially very informative, but they are complicated, and we believe that proper mathematical modelling will be required to explain these observations. Such an analysis is beyond the scope of the current manuscript, which is already complicated. Importantly, none of these findings invalidate or change any of the main conclusions of our current manuscript.

Dr. Jordan W Raff
University of Oxford
Sir William Dunn School of Pathology
South Parks Road
Oxford OX1 3RE
United Kingdom

17th Oct 2023

Re: EMBOJ-2023-115516

Rising Cdk/Cyclin levels promote, and then suppress, centrosome growth to help set centrosome size

Dear Jordan,

Thank you again for submitting your revised Review Commons manuscript on regulation of PCM accumulation at centrioles to The EMBO Journal. I am very sorry for the undue delay in getting back to you with a decision. Based on the potential interest expressed by the referees, I had chosen to contact them again, and received somewhat equivocal feedback that required further consultations with the referees and within our team. As you will see from the referee reports copied below, referees 1 and 3 were largely satisfied with the revisions in response to their Review Commons reports. However, referee 2 did retain a number of well-taken reservations, which during the follow-up consultations were also in part shared by referee 1. Key issues are that there is still no direct evidence for CDK phosphorylation of the key targets, and that the functionality of some of the employed reporter constructs has been insufficiently validated. I tend to agree with the referee that these points are central to the overall message of the paper, and therefore unfortunately conclude that without following-up on these issues, the study is not a sufficiently compelling candidate for EMBO Journal publication.

That said, with the overall interest of the work and the revisions that have already gone into it, the study would appear to be well-suited for publication in our sister journal EMBO Reports, in light of their focus on interesting key findings that do not necessarily need to be fully mechanistically fleshed out yet. I have therefore discussed it with my EMBO Reports counterpart, Dr. Deniz Senyilmaz Tiebe, who would indeed express interest in publishing this work without further experimental revisions. What she would in essence need would be a point-by-point response to the second round of referee comments (below) and textual addressing of all remaining concerns - in particular

- toning down the phrases regarding the direct phosphorylation of Spd-2 by Cdk (referees #1 and #2).
- including the mentioned S phase arrest experiment requested by referee #2 (e.g. in the EV or Appendix sections) and discussing the results
- discussing the caveats regarding the used reporters to monitor centrosomal dynamics in the manuscript.

Should you be interested in this option, please simply transfer the manuscript using the link at the bottom of this email, and Deniz would then provide you with an "official" revision invitation and formatting guidelines.

I am sorry that I could not be more positive regarding EMBO Journal publication in this case, but very much hope that you will find this opportunity for transferring and publishing without additional experiments worthwhile.

With kind regards,

Hartmut

Referee #1:

I have reviewed this study for Review Commons. Below is my general summary (largely based on my previous review) followed by specific points related to the new, revised manuscript.

The manuscript by Wong et al. investigates how cells regulate the increase in the size of the centrosomes (more specifically the

size of the pericentriolar material or PCM) that occurs during preparation for mitosis. They use the *Drosophila* syncytial embryo as a model, focusing on nuclear cycles 11-13, during which cell cycle progression gradually slows. The authors find that centrosomes grow to a consistent size at each cycle by adjusting to the slowed cell cycle, reducing the growth rate and increasing the growth period. This adjustment is proposed to be regulated by the Cdk/Cyclin cell cycle oscillator. Curiously, Cdk/Cyclin activity seems to both promote and inhibit the increase in centrosome size, depending on whether its activity is moderate or very high, respectively. Both effects are proposed to depend on the phosphorylation of centrosome proteins by Cdk/Cyclin.

This is an interesting manuscript that reaches somewhat different conclusions regarding centrosome size control when compared to previous studies in other organisms. In particular, work in *C. elegans* has proposed that centrosome growth regulation is controlled by the limited cytoplasmic availability of PCM building blocks, whereas the current study proposes a different model based on the activity of a cell cycle oscillator. The model system and approaches are well presented and the data is of good quality. The authors monitor a large number of centrosome markers, each with detailed quantifications of intensity and distribution over time during the different cycles. They also employ two different ways of quantifying centrosome size with similar results, making their quantifications more robust. While the authors include phospho-mutants in their analyses that presumably cannot be phosphorylated by Cdk/Cyclin, the study is largely descriptive. Still, the authors present interesting observations and propose the "oscillator model" an alternative to the "limited availability model" for the regulation of centrosome size, and perhaps that of other organelles. This study expands cell biologists' understanding of organellar size control.

The authors have clarified and addressed most of my concerns. Two minor issues remain:

1) Author rebuttal: "3. The Reviewer is puzzled as to how we conclude that Cdk/Cyclins phosphorylate Spd-2 and Cnn at all the potential Cdk/Cyclin phosphorylation sites we mutate in our study."

My original comment referred to the repeatedly stated claim in the original manuscript that Cdk DIRECTLY phosphorylates Spd-2 and Cnn at the mutated sites. Direct phosphorylation can only be concluded based on *in vitro* assays using purified kinase and substrate protein. While several of these statements have been removed in the revised manuscript, two such statements are still present and need to be rephrased (p.13, bottom; p.19, middle paragraph). "Cdk-dependent phosphorylation" would be an example of an appropriate description.

2) Inverse-linear relationship: I am not a mathematician, but I am still certain that the relationships shown in Fig. S5, contrary to the authors' description in results and rebuttal letter, are not "inverse-linear". They are "inverse" relationships. An "inverse-linear" relationship would be depicted by a line graph.

Referee #2:

In my comments on the original version of this manuscript I had expressed concerns about the strength of the evidence for the authors' model as well as the underlying methodology. I also found the manuscript difficult to follow in parts. In preparing this revision the authors have sought to address all of these concerns. However, I remain not fully convinced on the major points which I believe are central to the overall message of the paper. I therefore cannot support publication at this stage.

Main points

First and critically, there is the strength of the evidence for the proposed dual function of Cdk in both promoting and inhibiting centrosomal protein accumulation. I find it problematic that there is no experimental evidence presented in this manuscript that Cdk indeed phosphorylates the sites they mutate on Spd-2 and Cnn and there is only circumstantial evidence in the literature that they are phosphorylated at all. To their credit the authors themselves acknowledge this, but I find the authors' frequent use of words like 'appears to' troubling when discussing core aspects of their paper.

In my original review I had suggested the authors test their model by arresting embryos in S phase and mitosis and examining the effects on recruitment of PCM scaffold/client proteins. Based on the rebuttal this apparently has been done, but the results do not appear to be consistent with their predictions. Given the centrality of Cdk function to their model I believe this data absolutely should be presented and discussed here.

Second, there are the tools used to examine centrosomal protein dynamics, which are a mix of endogenously tagged reporters and GFP transgenes expressed under heterologous promoters of unknown functionality. I find the authors' argument based on different proteins expressed using different promoters exhibiting similar recruitment dynamics not convincing. A better argument is that, for Spd-2 at least, endogenous tagging yields similar recruitment kinetics as the original transgenic line. That the fluorescent reporters employed accurately reflect centrosomal protein dynamics is clearly critical to the models developed by the authors. This is therefore not a minor point, and must be better supported.

In this regard both reviewer 1 and I found it troubling that different components of the gamma-tubulin complex appear to exhibit different centrosomal distribution and recruitment dynamics. I perfectly understand that there are different gamma-tubulin

complexes, not all of which contain Grip128. However, this does not explain why that protein would exhibit a broader localization at centrosomes than gamma-tubulin itself, since all gamma-tubulin complexes (gamma-TuSC as well as TuRC) should contain gamma-tubulin, yet according to the authors' data there is supposedly a domain at the centrosome periphery occupied by a TuRC component, Grip128, but not gamma-tubulin. This makes no sense, unless the reporters (specifically the gamma-tubulin transgene) do not in fact accurately report on those proteins' true localization.

Finally, in response to reviewer 1's suggestion, the authors now examine centrosome function, specifically their microtubule organizing capacity. While certainly a worthwhile extension of the authors' work on the recruitment dynamics of PCM client proteins, I do not see this as essential to the core message of the paper. I do not, however, believe that the newly provided data adequately supports the authors' contention that centrosomal microtubule organizing capacity declines just prior to NEBD, in lockstep with the decline in centrosomal signal for key PCM client proteins they observe. Considering many of these proteins function in promoting microtubule nucleation or stability, it is perfectly possible that this is the case. However, one cannot make this point by monitoring the recruitment dynamics of yet another centrosomal MAP, Jupiter, not microtubules themselves.

Minor criticisms

p4 "As described above, Spd-2, Polo and Cnn, cooperate to guide the assembly of the mitotic PCM scaffold in flies. We therefore refer to these proteins as "scaffold" proteins, to distinguish them from PCM "client" proteins that interact with the scaffold, but which are not essential for scaffold assembly."

Even with this clarification I still see this as a problematic use of the word 'scaffold'. By that definition even Cdk could be described as a scaffold protein. Why not allow for a third term in addition to scaffold and client, 'regulator', which to me perfectly encapsulates the function of a protein like Polo?

p6 "Thus, in our experiments, centrosome "size" is defined as the amount of protein recruited, measured by centrosomal fluorescence intensity."

While there may be reasons for the authors' choice to focus on integrated fluorescence intensity, not volume/area, the preceding sentence does not provide any. 'Thus' is therefore not appropriate here.

p13 "Moreover, the subtle difference in the behaviour of Spd-2-NG in WT and CycB1/2 embryos was reproducible (Figure S9)." At first glance it looks like the authors here repeated the same experiment and got the same result, which seems superfluous. Only upon closer inspection does it become clear that the experiment in Fig S9 was performed with an endogenously tagged Spd-2 strain. The authors might want to indicate this in the text.

p14 "Importantly, the accumulation of Spd-2 and Cnn at centrosomes is sensitive to the genetic dosage of each protein, so the centrosomal levels of Spd-2 and Cnn are reduced in Spd-21/2 and Cnn1/2 embryos, respectively (Conduit et al, 2010) (S.S.W., unpublished).

I agree this is important. However, the authors should then include the evidence for Spd-2 in the manuscript, not leave it for another publication as the authors apparently intend to do.

p17 "Clearly more work is required to resolve this paradox <>, but it is important to note that the early Drosophila embryo is sensitive to the cytoplasmic levels of the key scaffolding proteins, so reducing Spd-2 or Cnn levels reduces mitotic centrosome size (Conduit et al, 2010) (Figure 6). It is just that in the fly system no scaffolding component seems to be sufficiently depleted from the cytoplasm to significantly limit centrosome growth as centrosome numbers increase."

It might be worth noting that in *C. elegans* early embryos unlike in later stage Drosophila syncytial embryos there is no ongoing centrosomal protein expression. Thus a limited protein pool is distributed amongst more and more centrosomes, while in flies new protein synthesis can compensate.

Referee #3:

All the points raised by me (reviewer 3) were completely answered in the revised manuscript.

*** As a service to authors, The EMBO Journal offers the possibility to directly transfer declined manuscripts to another EMBO Press title (EMBO Reports, EMBO Molecular Medicine, Molecular Systems Biology) or to the open access journal Life Science Alliance launched in partnership between EMBO Press, Rockefeller University Press and Cold Spring Harbor Laboratory Press. The full manuscript (including reviewer comments, where applicable and if chosen) will be automatically forwarded to the receiving journal, to allow for fast handling and a prompt decision on your manuscript. For more details of this service, and to transfer your manuscript to another EMBO title please follow this link:

Link Not Available

University of Oxford, South Parks Road, Oxford OX1 3RE

6th November, 2023

Dear Hartmut,

Re: #EMBOJ-2023-115516

Thanks for getting back to me about the manuscript. Obviously, we were disappointed with this outcome, but I appreciate that you and your team have put a lot of effort into trying to understand the paper and to identify and assess the key issues. As we briefly discussed, we believe we can address the two remaining concerns that you share with Reviewer #2 and highlight in your letter. I appreciate that in view of this you will reconsider the manuscript at EMBO Journal. I discuss these two points below and then provide a full point-by-point response to the Reviewer's new comments.

1. I deal with the Reviewer's second point first, as this is the most important. The Reviewer believes our reporter-constructs are not behaving properly, which, if correct, would clearly undermine our entire dataset. In light of the Reviewer's new comments we now understand the source of these concerns and can see that they arise due to a misunderstanding in presentation.

The reviewer is understandably puzzled about the centrosomal distribution of γ -tubulin and Grip128 reported in Figure S1 (which measures the area occupied by each protein). They note from the axes of the graphs that Grip128 seems to spread over an area of $\sim 4\mu\text{m}$, but γ -tubulin is only spread over an area of $\sim 1\text{-}2\mu\text{m}$. As it is impossible for Grip128 to occupy an area without γ -tubulin, they conclude that the γ -tubulin reporter must be faulty. Thus, our whole paper is flawed, as the faulty γ -tubulin reporter behaves pretty much like all the other reporters.

To explain what is going on here, I show in **Reviewer Figure 1** images of the Grip128 and Grip75 reporters—both CRISPR knock-ins of GFP at the endogenous locus (Tovey et al., *JCB*, 2021).

Reviewer Figure 1.

Left panels show images from time-lapse movies of embryos showing the centrosomal distribution of Grip75-GFP and Grip128-GFP. Right panels show how these images are computationally processed by our analysis pipeline (using the very common Otsu thresholding tool) to select the centrosome area for analysis (detected centrosomes shown in dark).

If you look at a still from a movie of each embryo (left images) you can see that their centrosomal localisation looks very similar. This is also true for γ -tubulin and all the Grip proteins we show in Figure 1A (and this very similar distribution explains, at least in part, why we didn't

initially understand the Reviewer's concern here). But, the Grip128-GFP reporter is actually a lot dimmer than the Grip75-GFP reporter. As a result, the Otsu thresholding function of our automated analysis pipeline segments a much larger area for Grip128 (right images). This has two consequences: (1) The computationally measured area of Grip128 is much larger than Grip75 (note y-axes of Figure S1C); (2) The change in centrosome "area" through the nuclear cycle is obvious for Grip71, but not for Grip128—as the computer thresholds such a large area for Grip128 that the area doesn't change much during the cycle (Figure 1D). Thus, the distribution and fluorescence intensity profiles of γ -tubulin and all the Grips look very similar (Figure 1A,C,D) (as we would expect if these GFP-fusions accurately report the distribution of their cognate proteins), but the area profiles of the dim proteins (Grip128 and Grip75) look quite different (Figure S1C) (due to the problem of accurately thresholding proteins with a low signal/noise ratio). We now explain this point in more detail in the Materials and Methods (and point this out more clearly in the Figure S1 Legend).

We apologise for not understanding this point well-enough to address it properly in our first response. All the reporters we use here are previously-validated and published reagents, and many are well-used in the field, so we didn't quite catch the significance of this point. This is a shame, as this misunderstanding has, understandably, led the Reviewer to question the validity of all our data.

2. The Reviewer is also concerned that we still have no evidence that Cdk/Cyclins *directly* phosphorylate key targets. While this is true, it is essentially impossible to prove that a specific kinase directly phosphorylates a specific residue in a living cell. If we abolish all Cdk activity and show that a specific site is no longer phosphorylated, this does not prove direct phosphorylation, as Cdk regulates the activity of so many other kinases. If we show that Cdk/Cyclin can phosphorylate a specific residue *in vitro*, this does not prove that Cdk phosphorylates that residue *in vivo*. If we show that Cdk/Cyclin does not detectably phosphorylate a specific residue *in vitro*, this does not prove that Cdk cannot phosphorylate that residue *in vivo* (e.g., both Spd-2 and Cnn are phosphorylated at certain sites only when recruited to centrosomes— Alvarez-Rodrigo et al., *eLife*, 2019, Figure 1A,B).

We would be willing to try an *in vitro* phosphorylation experiment if you feel it is essential, but it is expensive and quite time consuming and, as explained above, it cannot prove or disprove this point whatever the result.

Perhaps most importantly, this point of direct phosphorylation is not crucial to any of our main conclusions. The prevailing model for more than a decade has been that centrosome size is set by a limiting component (Decker et al., *Curr. Biol.*, 2011). We show that this is not correct in fly embryos. Instead, our data suggests that the cell cycle oscillator (CCO) sets centrosome size by reciprocally influencing the centrosome growth rate and centrosome growth period. If the cell cycle is short, the centrosomes grow quickly, but for a short period; if the cell cycle is long, they grow more slowly, but for a longer period; as a result, centrosomes grow to roughly the same size no matter how long the cycle (Figure 1, Figure 3 and Figure S5). We provide direct support for this hypothesis by showing that altering CCO activity (by halving the dose of Cyclin B) reciprocally alters the centrosome growth rate and period (Figure 4). This is a novel mechanism that, to our knowledge, has not been proposed before. We believe this warrants publication in EMBO J., even if we cannot definitively prove that Cdk/Cyclin is directly responsible for phosphorylating any of the centrosome proteins *in vivo*. In reality this complex phenomenon is likely to be regulated by multiple cell-cycle regulated phosphorylations driven by multiple cell cycle kinases, as we speculate in the Discussion.

We have now adjusted the title and abstract to more clearly focus on this key observation.

Point-by-point Response

Reviewer #1

The Reviewer was largely satisfied with our response to their concerns but two minor issues remained.

1. The Reviewer thought that in two places we still claimed that our data showed that Cdk directly phosphorylates Spd-2 and Cnn at the mutated sites (p13, bottom; p19, middle paragraph). We apologise if this is the case, although in neither of these places could we find a claim of direct phosphorylation that was not accompanied by a qualifier indicating that this was a “potential” direct phosphorylation, or that direct phosphorylation was “suggested”. Nevertheless, this is an important point and we have toned-down and/or clarified any statements regarding direct phosphorylation.

2. The Reviewer again questioned our use of the term “inverse-linear” to describe the fitted-curves we show in Figure S5. We have consulted again with our mathematicians on this and, although these curves are definitely “inverse-linear”, it seems the term may be somewhat ambiguous. Thus, as the Reviewer requests, and to avoid any confusion, we now simply state that they are “inverse”.

Reviewer #2

The Reviewer remained unconvinced by several of our arguments.

Main points:

1A. The Reviewer finds the lack of direct experimental evidence that Cdk phosphorylates any of the sites we mutate problematic. They state that there is only circumstantial evidence in the literature suggesting that these proteins are phosphorylated at all, and they find it troubling that we often use qualifying language (such as “appears to”) when discussing core aspects of our findings. The Reviewer is incorrect about the circumstantial evidence for phosphorylation, as we have previously shown that both Spd-2 and Cnn are phosphorylated specifically at centrosomes in fly embryos (see Figure 1B, Alvarez-Rodrigo et al., *eLife*, 2019). We agree, however, that we have not proved that Spd-2 or Cnn are directly phosphorylated by Cdk/Cyclin *in vivo*. We discuss this in point 2 of our general response, above. As we mention there, it is essentially impossible to prove direct phosphorylation *in vivo*, so we feel our use of qualifying language on this point is appropriate.

1B. The Reviewer notes that in response to their original request that we test our model by arresting embryos in S- and M-phase, we state that we have done this analysis, but the results are complicated, and we prefer to present them in a follow-up paper where we can explain them within the context of a mathematical model. The Reviewer concludes that these results are inconsistent with our predictions, and ask again that this data be presented here. I found this point upsetting, as it implies the Reviewer doesn’t believe our stated reasons for not showing this data and thinks instead that we are trying to hide results that are inconsistent with our model. This is not the case.

To elaborate more fully, we did not want to show this data here partly because it is complicated, but also because it is not very useful in the context of this paper. This is because nothing we present in this paper allows us to “predict” what would happen to these proteins in S- or M-phase arrested embryos. Figure 9 is a “schematic summary” (and is labelled as such), but it is not a model that can make predictions. In embryos arrested in M-phase, centrosomal Cnn-levels continue to increase indefinitely, while the other proteins quickly stabilise at a low steady-state level. *A priori*, this latter behaviour probably seems consistent with the cell-cycle regulation we are proposing, but we have no way to formally “predict” that this is the case. The behaviour of Cnn seems a bit odd (why does it keep growing, rather than reaching a steady-state level like the other proteins?), but again there is no way to formally tell whether this behaviour is inconsistent with the ideas we propose here.

The situation is even more complicated in embryos arrested in S-phase (by the depletion of Cyclins A, B and B3) (**Reviewer Figure 2**). Although no further rounds of mitosis occur, some

centrosomes can continue to duplicate, but they now do so asynchronously (McClelland and O'Farrell, *Curr. Biol.*, 2008; Aydogan et al., *Cell*, 2020) (arrows in Figure 2 highlight duplicating centrosomes 45min after Cyclin depletion). The centrosomes in these embryos tend towards a steady-state size, but each centrosome behaves differently: some centrosomes don't divide at all (and these tend to maintain their steady-state size), while those that do divide initially form two smaller centrosomes often of different sizes (probably depending on whether they are formed by the old-mother or new-mother centrioles—this size difference is highlighted with *Magenta* and *Cyan* arrows, Figure 2). We cannot analyse these centrosomes as we analysed all the others in the current paper because we cannot align the embryos to NEB (which does not occur in these embryos) or to centrosome separation (which either doesn't occur, or occurs asynchronously). We have developed analysis pipelines to overcome this problem, but the data has to be presented in a different way and so is not easy to digest without a significant amount of explanation.

Reviewer Figure 2. Images show an embryo expressing GFP-Cnn that was injected with Cyclin A, B and B3 RNAi ($t=0$). After ~90mins the embryo stops cycling and the centrosomes pass through their final round of co-ordinated division ($t=100$ mins). Although there are no further rounds of nuclear division, many centrosomes can continue to duplicate, but they do so stochastically. Thus centrosome numbers continue to increase ($t=135$ mins). Duplicating centrosomes at $t=45$ min are highlighted with arrows, Magenta indicating the larger, likely older-mother, centrosome, Cyan indicating the smaller, likely younger-mother, centrosome.

Again, this behaviour is probably intuitively consistent with our cell-cycle regulation hypothesis but, on intuition alone, it is hard to be sure, as the behaviour is so complicated. Fortunately, we previously published a simple mathematical model of PCM-scaffold assembly (Wong et al., *EMBO J.*, 2022). This did not account for any of the cell-cycle regulation we describe in the current paper but, with this new data in hand, we are now expanding the model to incorporate this important new aspect. This mathematical model can make proper predictions, and a crucial test is whether it can predict the behaviours we observe in these cell cycle arrested embryos (which it largely can, although this work is in progress).

We hope you will agree that these data are too complicated to fit neatly into the current manuscript, and that they would actually add very little to it. The significance of these observations cannot be appreciated without the context of a proper model that allows one to predict how these proteins should behave in these arrested-embryos.

2. The Reviewer remained unconvinced that the GFP-fusions we use here accurately report the true localisation of the centrosomal protein's of interest. We now understand why this confusion arose, and address this in the first point of our general response, above.

3. The Reviewer questioned the relevance of the new data we presented on the MT-organising capacity of the centrosome (Figure 2), and they particularly criticised our use of Jupiter-mCherry as a marker of MTs, requesting that we directly label MTs instead. We agree that this data is not central to our core message, but it was directly requested by

Reviewer #1. It is clear that fusing either α - or β -tubulin to fluorescent proteins in fly embryos strongly interferes with their function (although they still label MTs). Jupiter (either fused to GFP or mCherry) is now widely accepted as the standard way to label MTs in flies (used in >160 publications) as it faithfully reports MT localisation, without detectably perturbing MT function.

Minor Criticisms:

P4. The Reviewer still objects to our using the term 'scaffold' to describe Polo, and suggests we use 'regulator' instead. We understand this point, but adopting this terminology creates a problem in presentation, as we would no longer be able to easily delineate our old data describing Polo, Spd-2 and Cnn (which we group as Scaffold) compared to the new data (which we group as Clients). The Figures are already complicated enough without having to further split the data into more groups. To resolve this issue we now explain that although Polo is better described as a 'regulator', which we agree is correct, for ease of presentation we present the old Polo data grouped with the old scaffolding data, even though Polo is not a scaffold protein.

P6. As requested, we have now re-worded this sentence.

P13. As requested, we now more clearly state that Figure S9 was performed with an endogenously tagged Spd-2 strain.

P14. As requested, we have now included data showing that halving the genetic dosage of Spd-2 reduces the amount of Spd-2 recruited to centrosomes (New Figure S10).

P17. The Reviewer thought it might be worth noting that in early worm embryos there is no ongoing centrosomal protein expression, so a limited protein pool is distributed amongst more and more centrosomes, whereas in a later stage syncytial fly embryo new protein synthesis can compensate. This statement is slightly ambiguous, so we may not have understood it properly. In both early worm and fly embryos there is very little transcription, so development largely relies on maternally provided proteins and mRNAs (so, although there is no transcription, new proteins are continuously being made in both systems). Thus, it would be incorrect to suggest that developing worm embryos use a limiting component mechanism to set centrosome size because they can't make any new centrosomal proteins.

Reviewer #3

This Reviewer felt that we had addressed their initial concerns in our revised manuscript.

We hope you will agree that our explanations and final changes have addressed your remaining concerns and that our manuscript will be suitable for publication at EMBO J.

With best wishes,

Professor Jordan Raff
César Milstein Chair of Cancer Biology

Dr. Jordan W Raff
University of Oxford
Sir William Dunn School of Pathology
South Parks Road
Oxford OX1 3RE
United Kingdom

13th Nov 2023

Re: EMBOJ-2023-115516R-Q
Rising Cdk/Cyclin levels promote, and then suppress, centrosome growth to help set centrosome size

Dear Jordan,

Thank you for your detailed response to the last round of reviews on your Review Commons transfer, and for asking us to reconsider our decision on the manuscript. I have now eventually had a chance to look at all points in detail, as well as to go back to the referee reports from the previous to rounds of review. In light of your well-taken responses, in particular to the possible technical caveats raised by referee 2, I have concluded that the manuscript should be suitable for EMBO Journal publication after all, pending additional modifications in a final round of revision. This would require further adaptations to text and presentation in order to better explain certain aspects that appear to have remained unclear even after the first revision. Regarding the request for direct proof of CDK phosphorylation of centrosomal targets like Spd-2 or Cnn, I appreciate that this may not be as crucial to the key message of the study as it had initially appeared; and that the role of the cell cycle oscillator is in principle strongly supported by the cyclin B dosage alteration experiments. I also realize that the previous finding of their phosphorylation on Ser/Thr-Pro motifs together with the effects of mutating such sites to Ala or Glu provide further plausibility for the model. Therefore, I agree that attempting additional in vitro phosphorylation experiments would likely not substantially strengthen the study.

In this light, I am happy to invite you to resubmit (via the link below) a final version of the manuscript, incorporating the modifications and additional supportive figures described in your response letter. In addition to the scientific revisions, please at this stage also adjust the manuscript according to our formatting guidelines and incorporate the following editorial points:

- Please download and complete our author checklist (link provided below).
- Please upload the manuscript text as a separate DOCX file, and all main Figures (as well as any Expanded View figures) as individual files with sufficient resolution/quality for production.
- "Supplementary information" at EMBO Press has been replaced by Expanded View and Appendix (see embopress.org/page/journal/14602075/authorguide#expandedview for further guidance). Therefore, please consider converting up to 5 of the current "SI" figures into Expanded View Figures (these would be properly type-set -thus requiring the legends to be in the main text- and viewable in the HTML version); while converting the remainder into Appendix Figures compiled in a single Appendix PDF, together with their legends and prefaced by a brief Table of Contents.
- Please include a dedicated "Data Availability" section at the end of the Material and Methods (see embopress.org/page/journal/14602075/authorguide#dataavailability for its format); should there no data deposition to public repositories linked to the study, this should still be stated as "This study includes no data deposited in external repositories."
- Please adjust the header and the format of the Disclosure and competing interests statement (next to the Acknowledgment section) as specified in our Guide to Authors - for details, see <https://www.embopress.org/competing-interests>
- As we are switching from a free-text author contribution statement towards a more formal statement based on Contributor Role Taxonomy (CRediT) terms, please specify each author's contribution(s) directly in the Author Information page of our submission system during upload of the final manuscript, and remove the current Author Contribution section in the text. See <https://casrai.org/credit/> for more information.
- Please consider revising the title, to better emphasize the key novelty of the concept of CCO controlling centrosome size, and do mention, in light of contrasting reports from different organisms, your model system. Please also avoid the use of punctuation, which tends to make titles a bit clunky. Happy to discuss possible alternatives!
- Please double-check to make sure to all relevant funding information mentioned in the manuscript is congruent with the info entered into our submission system.

- Finally, please provide suggestions for a short 'blurb' text prefacing and summing up the conceptual aspect of the study in two sentences (max. 250 characters), followed by 3-5 one-sentence 'bullet points' with brief factual statements of key results of the paper; they will form the basis of an editor-written 'Synopsis' accompanying the online version of the article. Please also upload a synopsis image (maybe based on Figure 9?), which can be used as a "visual title" for the synopsis section of your paper. The image should be in PNG or JPG format, and please make sure that it remains in the modest dimensions of (exactly) 550 pixels wide and 300-600 pixels high.

Additional information on preparing, formatting and uploading a revised manuscript can be found below and in our Guide to Authors. Your re-revised manuscript will still undergo pre-acceptance checks by our staff as well as figure and legend editing/checking by our data editors; so closest possible adherence to these guidelines shall greatly facilitate our editorial processing at the resubmission stage.

Thank you again for the opportunity to consider this work for The EMBO Journal, and I look forward to receiving your final revised version.

With kind regards,

Hartmut

9) Digital image enhancement is acceptable practice, as long as it accurately represents the original data and conforms to community standards. If a figure has been subjected to significant electronic manipulation, this must be clearly noted in the figure legend and/or the 'Materials and Methods' section. The editors reserve the right to request original versions of figures and the original images that were used to assemble the figure. Finally, we generally encourage uploading of numerical as well as gel/blot image source data; for details see: embopress.org/page/journal/14602075/authorguide#sourcedata

At EMBO Press, we ask authors to provide source data for the main manuscript figures. Our source data coordinator will contact you to discuss which figure panels we would need source data for and will also provide you with helpful tips on how to upload and organize the files.

Further information is available in our Guide For Authors:

In the interest of ensuring the conceptual advance provided by the work, we recommend submitting a revision within 3 months (11th Feb 2024). Please discuss the revision progress ahead of this time with the editor if you require more time to complete the revisions. Use the link below to submit your revision:

Link Not Available

All editorial and formatting issues were resolved by the authors.

Dr. Jordan W Raff
University of Oxford
Sir William Dunn School of Pathology
South Parks Road
Oxford OX1 3RE
United Kingdom

13th Dec 2023

Re: EMBOJ-2023-115516R1
Regulation of centrosome size by the cell-cycle oscillator in Drosophila embryos

Dear Jordan,

Thank you for submitting your final revised manuscript for our consideration. I am pleased to inform you that we have now accepted it for publication in The EMBO Journal.

With kind regards,

Hartmut
